# Good Classification Measures and How to Find Them

**Martijn Gösgens**
Eindhoven University of Technology
Eindhoven, The Netherlands
research@martijngosgens.nl

**Anton Zhiyanov**
Yandex Research, HSE University
Moscow, Russia
zhiyanovap@gmail.com

**Alexey Tikhonov**
Yandex
Berlin, Germany
altsoph@gmail.com

**Liudmila Prokhorenkova**
Yandex Research, HSE University, MIPT
Moscow, Russia
ostroumova-la@yandex.ru

## Abstract

Several performance measures can be used for evaluating classification results: accuracy, F-measure, and many others. Can we say that some of them are better than others, or, ideally, choose one measure that is best in all situations? To answer this question, we conduct a systematic analysis of classification performance measures: we formally define a list of desirable properties and theoretically analyze which measures satisfy which properties. We also prove an impossibility theorem: some desirable properties cannot be simultaneously satisfied. Finally, we propose a new family of measures satisfying all desirable properties except one. This family includes the Matthews Correlation Coefficient and a so-called Symmetric Balanced Accuracy that was not previously used in classification literature. We believe that our systematic approach gives an important tool to practitioners for adequately evaluating classification results.

## 1   Introduction

*Classification* is a classic machine learning task that is used in countless applications. To evaluate classification results, one has to compare the predicted labeling of a given set of elements with the actual (true) labeling. For this, *performance measures* are used, and there are many well-known ones like accuracy, F-measure, and so on [13, 15]. The fact that different measures behave differently is known throughout the literature [10, 13, 20, 21]. For instance, accuracy is known to be biased towards the majority class. Thus, different measures may lead to different evaluation results, and it is important to choose an appropriate measure. While there are attempts to compare performance measures and describe their properties [3, 6, 13, 14, 26, 28], the problem still lacks a systematic approach, and our paper aims at filling this gap.[1] Our research is particularly motivated by a recent paper [12] providing a systematic analysis of evaluation measures for the *clustering* task. We transfer many proposed properties to the classification problem and extend the research by adding more properties, new measures, and novel theoretical results.

To provide a systematic comparison of performance measures, we formally define a list of properties that are desirable across various classification tasks. The proposed properties can be applied both to binary and multiclass problems. Some properties are intuitive and straightforward, like symmetry, while others are more tricky. A particularly important property is called *constant baseline*. It requires a measure not to be biased towards particular predicted class sizes. For each measure and each

---

[1]We describe related research in detail in Appendix A.

35th Conference on Neural Information Processing Systems (NeurIPS 2021).

property, we formally prove or disprove that the property is satisfied. We believe that this analysis is essential for better understanding the differences between the performance measures.

Then, we analyze relations between different properties in the binary case and prove an impossibility theorem: it is impossible for a performance measure to be linearly transformable to a metric and simultaneously have the constant baseline property. This means that at least one of these properties has to be discarded. If we relax the set of properties by discarding the distance requirement, the remaining ones can be simultaneously satisfied. In fact, we propose a family of measures called Generalized Means (GM), satisfying all the properties except distance and generalizing the well-known Matthews Correlation Coefficient (CC). In addition to CC, this class also contains another intuitive measure that we name *Symmetric Balanced Accuracy*. To the best of our knowledge, this measure has not been previously used for classification evaluation.[2] If we instead discard the constant baseline (but keep its approximation), then the arccosine of CC is a measure satisfying all the properties.

We also demonstrate through a series of experiments that different performance measures can be inconsistent in various situations. We notice that measures having more desirable properties are usually more consistent with each other.

We hope that our research will motivate further studies analyzing the properties of performance measures for classification and other problems since there are still plenty of questions to be answered.

## 2 Performance measures for classification

In this section, we define measures that are commonly used for evaluating classification results. Classification problems can be divided into binary, multiclass, and multilabel. In this paper, we focus on *binary* and *multiclass* and leave multilabel for future research. There are several types of performance measures: *threshold* measures assume that predicted labels deterministically assign each element to a class (e.g., accuracy); *probability* measures assume that the predicted labels are soft and compare these probabilities with the actual outcome (e.g., the cross-entropy loss); *ranking* measures take into account the relative order of the predicted soft labels, i.e., quantify whether the elements belonging to a class have higher predicted probabilities compared to other elements (e.g., area under the ROC curve or average precision). Our research focuses on threshold measures.

Now we introduce notation needed to formally define binary and multiclass threshold measures.[3] Let $n > 0$ be the number of elements in the dataset and let $m \geq 2$ denote the number of classes. We assume that there is *true labeling* classifying elements into $m$ classes and also *predicted labeling*. Let $\mathcal{C}$ be the confusion matrix: each matrix element $c_{ij}$ denotes the number of elements with true label $i$ and predicted label $j$. For binary classification, $c_{11}$ is *true positive* (TP), $c_{00}$ is *true negative* (TN), $c_{10}$ is *false negative* (FN), and $c_{01}$ is *false positive* (FP). We use the notation $a_i = \sum_{j=0}^{m-1} c_{ij}, b_i = \sum_{j=0}^{m-1} c_{ji}$ for the sizes of $i$-th class in the true and predicted labelings, respectively. Finally, we denote classification measures by $M(\mathcal{C})$ or $M(A, B)$, where $A$ and $B$ are true and predicted labelings, and write $M(c_{11}, c_{10}, c_{01}, c_{00})$ for binary ones.

Table 1 (above the line) lists several widely used classification measures. The most well-known is *accuracy* which is the fraction of correctly classified elements. Accuracy is known to be biased towards the majority class, so it is not appropriate for unbalanced problems. To overcome this, *Balanced Accuracy* re-weights the terms to treat all classes equally. *Cohen's Kappa* uses a different approach to overcome this bias: it corrects the number of correctly classified samples by the expected value obtained by a random classifier [5]. *Matthews Correlation Coefficient* is the Pearson correlation coefficient between true and predicted labelings for binary classification [11]. For the multiclass case, covariance is computed for each class, and the obtained values are averaged before computing the correlation coefficient. Finally, *Confusion Entropy* computes the entropy of the misclassification distribution for each class and combines the obtained values, see Table 1 and [30] for the details.[4]

Some measures are exclusively defined for binary classification. In this case, the classes are often referred to as 'positive' and 'negative'. *Jaccard* measures the fraction of correctly detected positive

---

[2]For clustering evaluation, there is an analog known as *Sokal&Sneath's measure* [12].

[3]For convenience, we list the notation used in the paper in Table 7 in Appendix.

[4]There can be cases when a class is not present in the predicted labels. Then, some measures may contain division by zero. A proper way to fill in such singularities is discussed in Appendix B.

Table 1: Commonly used (above the line) and novel (below the line) validation measures

| | Binary | Multiclass |
|---|---|---|
| F-measure ($F_\beta$) | $\frac{(1+\beta^2)\cdot c_{11}}{(1+\beta^2)\cdot c_{11}+\beta^2\cdot c_{10}+c_{01}}$ | micro / macro / weighted |
| Jaccard (J) | $\frac{c_{11}}{c_{11}+c_{10}+c_{01}}$ | micro / macro / weighted |
| Matthews Coefficient (CC) | $\frac{c_{11}c_{00}-c_{01}c_{10}}{\sqrt{b_1\cdot a_1\cdot b_0\cdot a_0}}$ | $\frac{n\sum_{i=0}^{m-1}c_{ii}-\sum_{i=0}^{m-1}b_i a_i}{\sqrt{\left(n^2-\sum_{i=0}^{m-1}b_i^2\right)\left(n^2-\sum_{i=0}^{m-1}a_i^2\right)}}$ |
| Accuracy (Acc) | | $\frac{\sum_{i=0}^{m-1}c_{ii}}{n}$ |
| Balanced Accuracy (BA) | | $\frac{1}{m}\sum_{i=0}^{m-1}\frac{c_{ii}}{a_i}$ |
| Cohen's Kappa ($\kappa$) | | $\frac{n\sum_{i=0}^{m-1}c_{ii}-\sum_{i=0}^{m-1}a_i b_i}{n^2-\sum_{i=0}^{m-1}a_i b_i}$ |
| Confusion Entropy (CE) | | $-\frac{1}{2n}\sum_{i,j:i\neq j}\left(c_{ji}\log_{2m-2}\frac{c_{ji}}{a_j+b_j}+c_{ij}\log_{2m-2}\frac{c_{ij}}{a_j+b_j}\right)$ |
| Symmetric Balanced Accuracy (SBA) | | $\frac{1}{2m}\sum_{i=0}^{m-1}\left(\frac{c_{ii}}{a_i}+\frac{c_{ii}}{b_i}\right)$ |
| Generalized Means (GM) | $\frac{n\,c_{11}-a_1 b_1}{\sqrt[r]{\frac{1}{2}\left(a_1^r a_0^r+b_1^r b_0^r\right)}}$ | micro / macro / weighted |
| Correlation Distance (CD) | | $\frac{1}{\pi}\arccos(\text{CC})$ |

examples among all positive ones (both in true and predicted labelings). *F-measure* is the (possibly weighted) harmonic mean of Recall ($c_{11}/a_1$) and Precision ($c_{11}/b_1$). For measures that do not have a natural multiclass variant, there are several universal extensions obtained via *averaging* the results for $m$ one-vs-all binary classifications [17]. For each one-vs-all classification, a particular class $i$ is considered positive while all other classes are grouped to a negative class.

*Micro averaging* sums up all binary confusion matrices corresponding to $m$ one-vs-all classifications. Formally, it sets true positive as $\sum_{i=0}^{m-1}c_{ii}$, false negative and false positive as $n-\sum_{i=0}^{m-1}c_{ii}$, true negative as $(m-2)n+\sum_{i=0}^{m-1}c_{ii}$. Then, a given binary measure is applied to the obtained matrix.

*Macro averaging* computes the measure values for $m$ binary classification sub-problems and then averages the results: $\frac{1}{m}\sum_{i=0}^{m-1}M(c_{ii},a_i-c_{ii},b_i-c_{ii},n-a_i-b_i+c_{ii})$, where $M(\cdot)$ is a given binary measure. Note that macro averaging gives equal weights to all one-vs-all binary classifications.

In contrast, *weighted averaging* weights one-vs-all binary classifications according to the sizes of the corresponding classes: $\frac{1}{n}\sum_{i=0}^{m-1}a_i\cdot M(c_{ii},a_i-c_{ii},b_i-c_{ii},n-a_i-b_i+c_{ii})$.

# 3 Properties of validation measures

As clearly seen from the above discussion, there are many options for classification validation. In this section, we propose a formal approach that allows for a better understanding the differences between the measures and for making an informed decision among them for a particular application. For this, we propose properties of validation measures that can be useful across various applications and formally check which measures satisfy which properties. In this regard, we follow the approach proposed in [12] for comparing validation measures for *clustering* tasks.

First, we observe that some theoretical results from [12] are related to *binary* classification measures. Indeed, a popular subclass of clustering validation measures are *pair-counting* ones. Such measures are defined in terms of the values $N_{11},N_{10},N_{01},N_{00}$ that essentially define a confusion matrix for binary classification on *element pairs*. Thus, replacing $N_{ij}$ in pair-counting clustering measures by $c_{ij}$, results in *binary* classification measures. We refer to Table 8 in Appendix B for the correspondence of some classification and clustering measures. In particular, Accuracy is equivalent to Rand, while Cohen's Kappa corresponds to Adjusted Rand. This equivalence allows us to transfer some of the results from [12] to the context of binary classification. However, an important contribution of our work is the extension of the properties and analysis to the multiclass case. We also prove an impossibility theorem stating that some of the desirable properties cannot be simultaneously satisfied and develop a new family of measures having all properties except one.

Table 2: Properties of validation measures and averagings, ✓/✗ indicates that property is satisfied only in binary case

| Measure | Max | Min | CSym | Sym | Dist | Mon | SMon | CB | ACB |
|---|---|---|---|---|---|---|---|---|---|
| $F_1$ (binary) | ✓ | ✗ | ✗ | ✓ | ✗ | ✓ | ✗ | ✗ | ✗ |
| J (binary) | ✓ | ✗ | ✗ | ✓ | ✓ | ✓ | ✗ | ✗ | ✗ |
| CC | ✓ | ✓/✗ | ✓ | ✓ | ✗ | ✓/✗ | ✓/✗ | ✓ | ✓ |
| Acc | ✓ | ✓ | ✓ | ✓ | ✓ | ✓ | ✓ | ✗ | ✗ |
| BA | ✓ | ✓ | ✓ | ✗ | ✗ | ✓ | ✓ | ✓ | ✓ |
| $\kappa$ | ✓ | ✗ | ✓ | ✓ | ✗ | ✓/✗ | ✗ | ✓ | ✓ |
| CE | ✓ | ✗ | ✓ | ✓ | ✗ | ✗ | ✗ | ✗ | ✗ |
| SBA | ✓ | ✓ | ✓ | ✓ | ✗ | ✓ | ✓ | ✓ | ✓ |
| GM (binary) | ✓ | ✓ | ✓ | ✓ | ✗ | ✓ | ✓ | ✓ | ✓ |
| CD | ✓ | ✓/✗ | ✓ | ✓ | ✓ | ✓/✗ | ✓/✗ | ✗ | ✓ |
| Preserving properties by various averaging types | | | | | | | | | |
| Micro | ✓ | ✗ | ✓ | ✓ | ✓ | ✓ | ✗ | ✗ | ✗ |
| Macro | ✓ | ✗ | ✓ | ✓ | ✓ | ✓ | ✗ | ✓ | ✓ |
| Weighted | ✓ | ✗ | ✓ | ✗ | ✗ | ✓ | ✗ | ✓ | ✓ |

Similarly to [12], we note that all the discussed properties are invariant under linear transformations and interchanging true and predicted labelings. Hence, we may restrict to measures for which higher values indicate higher similarity between classifications.

Table 2 summarizes our findings: for each measure, we mathematically prove or disprove each desirable property. Further in this section, we refer only to known measures (above the line), while the remaining ones will be defined and analyzed in Section 4. In addition to individual measures, we also analyze the properties of micro, macro, and weighted multiclass averagings: for each averaging, we analyze whether it preserves a given property, assuming the binary classification measure satisfies it. All the proofs can be found in Appendix C. Let us now define and motivate each property.

### 3.1 Maximal and minimal agreement

These properties make the upper and lower range of a performance measure interpretable. The *maximal agreement* property requires the measure to have an upper bound that is only achieved when the compared labelings are identical.

**Definition 1.** *We say that a measure $M$ satisfies* maximal agreement *if there exists a constant $c_{\max}$ such that for all $\mathcal{C}$, $M(\mathcal{C}) \leq c_{\max}$ with equality iff $\mathcal{C}$ is diagonal.*

Also, for a given true labeling, there are several "worst" predictions, i.e., labelings that are wrong everywhere. This leads to the following property.

**Definition 2.** *We say that a measure $M$ satisfies* minimal agreement *if there exists a constant $c_{\min}$ such that for all $\mathcal{C}$, $M(\mathcal{C}) \geq c_{\min}$ with equality iff the diagonal of $\mathcal{C}$ is zero, i.e., $c_{ii} = 0$ for all $i$.*

These properties allow for an easy and intuitive interpretation of the measure's values. While all of the measures in Table 2 do satisfy maximal agreement, there are popular measures such as Recall ($c_{11}/a_1$) and Precision ($c_{11}/b_1$) that do not satisfy this property as the maximum can also be achieved when the compared classifications are not identical. For minimal agreement, many performance measures violate it. For example, Cohen's Kappa is obtained from accuracy by subtracting the expected value of accuracy and normalizing the result. As a result of the particular normalization used, it has minimal value $- \left( \sum_{i=0}^{m-1} a_i b_i \right) / \left( n^2 - \sum_{i=0}^{m-1} a_i b_i \right)$, which is clearly not constant.

If a binary measure satisfies maximal agreement, then its multiclass variant obtained via micro, macro, or weighted averaging also satisfies this property as each one-vs-all binary classification agrees maximally. However, this does not hold for minimal agreement: though each one-vs-all binary classification will have zero true positives, the number of true negatives may still be positive.

## 3.2 Symmetry

**Definition 3.** *We say that a measure $M$ is* symmetric *if $M(\mathcal{C}) = M(\mathcal{C}^T)$ holds for all $\mathcal{C}$.*

In other words, we require symmetry with respect to interchanging predicted and true labels. This property is often desirable since similarity is usually understood as a symmetric concept. However, in some specific applications, there may be reasons to treat the true and predicted labelings differently and thus use an asymmetric measure. An example of an asymmetric measure is Balanced Accuracy.

Let us also introduce *class-symmetry*, i.e., invariance to permuting the classes.

**Definition 4.** *We say that a measure $M$ is* class-symmetric *if, for any permutation $\pi$ of the classes $\{1, \ldots, m\}$ and any confusion matrix $\mathcal{C}$, $M(\mathcal{C}) = M(\tilde{\mathcal{C}})$ holds, where $\tilde{\mathcal{C}}$ is given by $\tilde{c}_{ij} = c_{\pi(i), \pi(j)}$.*

Note that known multiclass measures are all class-symmetric, while in binary classification tasks, there can be an asymmetry between 'positive' and 'negative' classes. Examples of well-known class-asymmetric binary classification measures are Jaccard and $F_1$.

## 3.3 Distance

In some applications, it is desirable to have a distance interpretation of a measure: whenever a labeling $A$ is similar to $B$, while $B$ is similar to $C$, it should intuitively hold that $A$ is also somewhat similar to $C$. For instance, it can be the case that the *actual* labels are unknown, and the labeling $A$ is only an approximation of the truth. Then, we would want the similarity between predicted labels and $A$ to be not too different from the similarity between predicted and the actual true labels. This would be guaranteed if the measure is a distance.

**Definition 5.** *A measure has* distance *property if it can be linearly transformed to a metric distance.*

A function $d(A, B)$ is a metric distance if it is symmetric, nonnegative, equals zero only when $A = B$, and satisfies the triangle inequality $d(A, C) \leq d(A, B) + d(B, C)$. Note that the first requirement is equivalent to symmetry (Definition 3), while the second and third imply maximal agreement (Definition 1). Furthermore, note that if $d$ is a distance, then $c \cdot d$ is also a distance for any $c > 0$. Therefore, we can conclude that $M$ is a distance if and only if $M$ satisfies symmetry and maximal agreement while $c_{\max} - M(A, B)$ satisfies the triangle inequality.

While most measures cannot be linearly transformed to a distance, some measures do satisfy this property. For example, the Jaccard measure can be transformed to the Jaccard distance $1 - \mathrm{J}(A, B)$. Similarly, Accuracy can be transformed to a distance by $1 - \mathrm{Acc}(A, B)$.

## 3.4 Monotonicity

Monotonicity is one of the most important properties of a similarity measure: intuitively, changing one labeling such that it becomes more similar to the other ought to increase the similarity score. Then, to formalize monotonicity, we need to determine what changes make the classifications $A$ and $B$ more similar to each other. The simplest option is to take one element on which $A$ and $B$ disagree and resolve this disagreement.

**Definition 6.** *A measure $M$ is* monotone *if $M(\mathcal{C}) < M(\tilde{\mathcal{C}})$ for any confusion matrices $\mathcal{C}$ and $\tilde{\mathcal{C}}$ such that $\tilde{\mathcal{C}}$ is obtained from $\mathcal{C}$ by decrementing an off-diagonal entry $c_{ab}$ and incrementing $c_{aa}$ or $c_{bb}$ and none of the row- or column-sums of $\mathcal{C}$ equal $n$.*

The condition on $\mathcal{C}$ is equivalent to neither $A$ nor $B$ labeling all elements to the same class. We need this to prevent contradictions with the constant baseline property that will be defined in Section 3.5.

Definition 6 defines a partial ordering over confusion matrices with the same total number of elements. However, we can relax the latter restriction and obtain the following, stronger notion of monotonicity that defines a partial ordering across different numbers of elements.

**Definition 7.** *A measure $M$ is* strongly monotone *if $M(\mathcal{C}) < M(\tilde{\mathcal{C}})$ for any confusion matrices $\mathcal{C}$ and $\tilde{\mathcal{C}}$ such that $\tilde{\mathcal{C}}$ is obtained from $\mathcal{C}$ by either increasing a diagonal entry or decreasing an off-diagonal entry. Here we require that none of the row- or column-sums of $\mathcal{C}$ equal $n$ and that $\mathcal{C}$ and $\tilde{\mathcal{C}}$ are not simultaneously diagonal or zero-diagonal matrices.*

The last condition is needed since otherwise the definition would contradict the maximal (or minimal) agreement properties as $M(\mathcal{C}) = c_{\max} \geq M(\tilde{\mathcal{C}})$ holds when $\mathcal{C}$ is diagonal.

All measures in Table 2 except for CE and multiclass $\kappa$, CC and CD satisfy monotonicity from Definition 6. Strong monotonicity is violated by many measures: for instance, the widely used $F_1$, Jaccard and Cohen's Kappa do not satisfy this intuitive property.

### 3.5 Constant baseline

The constant baseline is perhaps the most important non-trivial property. On the one hand, it ensures that a measure is not biased towards labelings with particular class sizes $b_1, \ldots, b_m$. On the other hand, it also ensures some interpretability for 'mediocre' predictions.

Intuitively, if predicted labels are drawn at random and independently of the true labels, we would expect them to have a low similarity with the true labels. Then, if another prediction has a similarly low score, we can say that it is roughly as bad as a random guess. However, this is only possible when such random classifications achieve similar scores, independent of their class sizes. To formalize this, let $U(b_1, \ldots, b_m)$ denote the uniform distribution over labelings with class sizes $b_1, \ldots, b_m$. We say that the class sizes $b_1, \ldots, b_m$ are *unary* if $b_i = n$ for some $i \in \{1, \ldots, m\}$. That is, if all elements get classified to the same class, so that $U(b_1, \ldots, b_m)$ is a constant distribution.

**Definition 8.** *We say that a measure $M$ has a* constant baseline *property if there exists $c_{base}(m)$ that does not depend on $n$ but may depend on $m$, such that for any true labeling $A$ and non-unary class sizes $b_1, \ldots, b_m$, it holds that $\mathbb{E}_{B \sim U(b_1, \ldots, b_m)}[M(A, B)] = c_{base}(m)$.*

Note that we need to require the class sizes to be non-unary: if these class sizes are unary, we will have contradictions with maximal and minimal agreement when the class sizes of $A$ are also unary. Many popular measures such as $F_1$, Accuracy, and Jaccard do not have a constant baseline. Furthermore, some measures that do have a constant baseline were deliberately designed to have one. For example, Cohen's Kappa was obtained from accuracy by correcting it for chance. While our definition of the constant baseline does allow for a baseline $c_{\text{base}}(m)$ that depends on the number of classes $m$, some measures such as the Matthews Coefficient and Cohen's Kappa have a baseline that is constant w.r.t. $m$.

All of the measures that satisfy constant baseline turn out to be linear functions of $c_{ii}$ for fixed class sizes $a_1, \ldots, a_m$ and $b_1, \ldots, b_m$. For such measures, linearity of the expectation can be utilized to easily compute the baseline by substituting the expected values $\mathbb{E}_{B \sim U(b_1, \ldots, b_m)}[c_{ii}] = \frac{a_i b_i}{n}$. Thus, we also propose the following relaxation of the constant baseline property.

**Definition 9.** *A measure $M$ is said to have an* approximate constant baseline *if there exists a function $c_{base}(m)$ that does not depend on $n$ but may depend on $m$ such that for any class sizes $a_1, \ldots, a_m$ and any non-unary $b_1, \ldots, b_m$, $M(\bar{\mathcal{C}}) = c_{base}(m)$, where $\bar{c}_{ij} = \frac{a_i b_j}{n}$.*

The advantage of this relaxation is that it allows us to non-linearly transform measures while still maintaining an approximate constant baseline. Take for example the Matthews Correlation Coefficient: it cannot be linearly transformed to a distance while the transformations $\text{CD}(A, B) = \frac{1}{\pi} \arccos(\text{CC}(A, B))$ and $\sqrt{2(1 - \text{CC}(A, B))}$ do yield distances. Because Correlation Coefficient has a constant baseline, these non-linear transformations have an approximate constant baseline, see Section 4 for more details.

As can be seen from Table 2, there is no measure satisfying all the properties. In particular, there is no measure having both distance and constant baseline. In the next section, we show why this is not a coincidence.

## 4 Impossibility theorem for classification

In this section, we focus on binary classification and more deeply analyze the relations between the properties discussed above. Unfortunately, it turns out that the properties introduced in the previous section cannot all be satisfied simultaneously.

**Theorem 1.** *There is no binary classification measure that simultaneously satisfies the monotonicity, distance, and constant baseline properties.*

*Proof.* Let $A$ be a labeling with a single positive and $n-1$ negatives. Let $B_1$ be a random labeling with a single positive and let $B_2$ be a random labeling with two positives. The constant baseline requires $\mathbb{E}[M(A, B_1)] = \mathbb{E}[M(A, B_2)]$, which gives

$$\frac{1}{n}c_{\max} + \frac{n-1}{n}M(0,1,1,n-2) = \frac{2}{n}M(1,0,1,n-2) + \frac{n-2}{n}M(0,1,2,n-3),$$

which we rewrite to

$$2M(1,0,1,n-2) - c_{\max} = (n-1)M(0,1,1,n-2) - (n-2)M(0,1,2,n-3). \tag{1}$$

Now, we consider a labeling $C$ with a single positive that does not coincide with the positive of $A$ and a labeling $B$ that has two positives which are the positives of $A$ and $C$. The triangle inequality tells us that

$$c_{\max} - M(0,1,1,n-2) \leq 2c_{\max} - M(1,1,0,n-2) - M(1,0,1,n-2) = 2(c_{\max} - M(1,1,0,n-2)),$$

where the last step follows from symmetry (implied by distance). This is rewritten to

$$2M(1,1,0,n-2) - c_{\max} \leq M(0,1,1,n-2). \tag{2}$$

Combining (1) and (2), we obtain

$$(n-1)M(0,1,1,n-2) - (n-2)M(0,1,2,n-3) \leq M(0,1,1,n-2).$$

We rewrite this to $M(0,1,1,n-2) \leq M(0,1,2,n-3)$, which clearly contradicts monotonicity. $\square$

Thus, we have to discard one of these properties. Obviously, discarding monotonicity would be highly undesirable since higher values would then not necessarily indicate higher similarity. For this reason, we analyze what happens if we discard either *distance* or *constant baseline*. All the results stated below are proven in Appendix D.

**Discarding distance**    Assuming some additional smoothness conditions that are, however, satisfied by all measures discussed in this paper, we prove the following result.

**Theorem 2.** *All binary measures that satisfy all properties except distance must be of the form*

$$s\left(\frac{a_0 a_1}{n^2}, \frac{b_0 b_1}{n^2}\right) \cdot \frac{n c_{11} - a_1 b_1}{n^2},$$

*where the normalization factor $s(a, b)$ needs to satisfy some additional properties listed in Theorem 3.*

This class of measures is quite wide and contains many unelegant measures. An interesting subclass can be obtained if we normalize by the generalized mean, i.e., take $s(a,b)^{-1} = (\frac{1}{2}a^r + \frac{1}{2}b^r)^{1/r}$.

**Definition 10.** *For $r \in \mathbb{R}$, we define* Generalized Means *measures as*

$$GM_r = \frac{n\, c_{11} - a_1 b_1}{\sqrt[r]{\frac{1}{2}\left(a_1^r a_0^r + b_1^r b_0^r\right)}}.$$

**Statement 1.** *For any $r \in \mathbb{R}$, the measure $GM_r$ satisfies all properties except for being a distance.*

We also show that the Generalized Means measures contain two interesting special cases.

**Statement 2.** *If $r \to 0$ (corresponding to the geometric mean), $GM_r(\mathcal{C}) \to CC(\mathcal{C})$.*
*If $r = -1$ (corresponding to the harmonic mean), $GM_{-1}(\mathcal{C}) = BA(\mathcal{C}) + BA(\mathcal{C}^\top) - 1$.*

Thus, for $r = -1$ Generalized Means is equivalent to the measure $\frac{1}{2}\left(BA(\mathcal{C}) + BA(\mathcal{C}^\top)\right)$ that we call *Symmetric Balanced Accuracy* (SBA). To the best of our knowledge, this measure has not been used in the classification literature. However, in the clustering literature, a similar measure is known as Sokal&Sneath's measure [1, 12]. Interestingly, SBA preserves its properties for the multiclass case.

**Statement 3.** *SBA satisfies all properties except for being a distance for any $m \geq 2$.*

**Discarding (exact) constant baseline**  Note that Theorem 1 only proves an impossibility for the *exact* constant baseline, but not the *approximate* constant baseline.

**Statement 4.** *The measures* $\mathrm{CD}(A, B) := \frac{1}{\pi} \arccos(\mathrm{CC}(A, B))$ *and* $\mathrm{CD}'(A, B) := \sqrt{2(1 - \mathrm{CC}(A, B))}$ *satisfy all properties except the exact constant baseline, but including the approximate constant baseline.*

Following [12], we call the measure $\frac{1}{\pi} \arccos(\mathrm{CC}(A, B))$ *Correlation Distance* (CD). We prove the following result (see Appendix D.2 for the details).

**Statement 5.** CD *approximates a constant baseline with one order of precision better than* $\mathrm{CD}'$.

Essentially, this is a consequence of the fact that the transformation $\frac{1}{\pi} \arccos(\mathrm{CC})$ is a symmetric function around the constant baseline $\mathrm{CC} = 0$ while $\sqrt{2(1 - \mathrm{CC})}$ is not. In more detail, we show that the leading error term of $\mathrm{CD}'$ is of the order $\mathbb{E}[\mathrm{CC}(A, B)^2]$ while the leading error term for CD is of the order $\mathbb{E}[\mathrm{CC}(A, B)^3]$. Currently, we are not aware of other distance measures for which the constant baseline is approximated up to the same order of precision as CD. We thus argue that for binary classification tasks where a distance interpretation is desirable, Correlation Distance is the most suitable measure.

## 5  Inconsistency of measures in practice

In this section, we conduct several experiments that demonstrate how often performance measures may disagree in practice in different scenarios. These experiments demonstrate the importance of the problem considered in this paper and show which measures are usually more consistent than others. For binary classification, we consider all measures from Table 1. For F-measure, we take $\beta = 1$, for Generalized Means, we consider $r = 1$. Recall that SBA and CC are also instances of GM with $r = -1$ and $r \to 0$, respectively. Furthermore, Jaccard is a monotone transformation of $F_1$, and CD is a monotone transformation of CC. Therefore, we omit CD and Jaccard from all inconsistency tables. The code for our experiments can be found on GitHub.[5]

### 5.1  Binary measures

**Distinguishing measures for small datasets**  First, we construct simple examples showing the inconsistency of all pairs of binary classification measures. We say that two measures $M_1$ and $M_2$ are consistent on a triplet of classifications $(A, B_1, B_2)$ if $M_1(A, B_1) * M_1(A, B_2)$ implies $M_2(A, B_1) * M_2(A, B_2)$, where $* \in \{>, <, =\}$. Otherwise, we say that the measures are inconsistent. We took $n \in \{2, 3, \ldots, 10\}$ and went through all the possible triplets $(A, B_1, B_2)$ of binary labelings of $n$ elements (we additionally require that all labelings contain both classes). For each triplet, we check which pairs of measures are inconsistent. We say that a pair of measures is indistinguishable for a given $n$ if it is consistent on all triplets.

Table 3 lists all measures that are indistinguishable for a given $n$. For instance, for $n = 2$, all measures are always consistent. For $n = 4$, we can distinguish Acc, $F_1$, and CE from other measures and each other. Interestingly, the remaining measures are those having the constant baseline property. Importantly, the most consistent measures are CC, SBA, and $\mathrm{GM}_1$ — these measures have the best properties according to our analysis. This supports our intuition that "good" measures agree with each other better than those having fewer desired properties. Additionally, in Appendix E.1, we list six triplets $(A, B_1, B_2)$ with $n = 10$ that discriminate all pairs of different measures.

Table 3: Indistinguishable measures

| $n$ | measures |
| --- | --- |
| 2 | [Acc, BA, $F_1$, $\kappa$, CE, $\mathrm{GM}_1$, CC, SBA] |
| 3 | [Acc, BA, $\kappa$, $\mathrm{GM}_1$, CC, SBA] |
| 4-5 | [BA, $\kappa$, $\mathrm{GM}_1$, CC, SBA] |
| 6-7 | [$\mathrm{GM}_1$, CC, SBA] |
| 8 | [CC, SBA] |
| 9-10 | — |

**Experiment within a weather forecasting service**  In this experiment, we aim at understanding whether the differences between measures may affect the decisions made while designing real systems. For this purpose, we conduct an experiment within the *Yandex.Weather* service.

---

[5] https://github.com/yandex-research/classification-measures

There is a model that predicts the presence/absence of precipitation at a particular location [18]. The prediction is made for 12 prediction intervals (*horizons*): from ten minutes to two hours. The original model returns the probability of precipitation, which can be converted to binary labels via a threshold. There are six thresholds used in this experiment, which lead to six different models. The measures were logged for 12 days. To sum up, for each threshold (model), each day, and each horizon, we have a confusion matrix that can be used to compute a performance measure.

For each pair of measures, we compute how often they are inconsistent according to the definition above. For this, we aggregate the results over all days and horizons. Table 4 shows that there are pairs of measures with substantial disagreement: e.g., accuracy and Balanced Accuracy almost always disagree. This can be explained by the fact that accuracy has a bias towards the majority class, so it prefers a higher

Table 4: Inconsistency of binary measures for rain prediction, %

|        | Acc  | BA   | $F_1$ | $\kappa$ | CE   | $GM_1$ | CC   | SBA  |
|--------|------|------|-------|----------|------|--------|------|------|
| Acc    | —    | 96.5 | 41.0  | 37.5     | 3.1  | 38.7   | 44.3 | 55.9 |
| BA     | 96.5 | —    | 55.6  | 58.9     | 99.7 | 57.7   | 52.0 | 40.4 |
| $F_1$  | 41.0 | 55.6 | —     | 3.3      | 44.2 | 2.2    | 3.4  | 15.0 |
| $\kappa$ | 37.5 | 58.9 | 3.3   | —        | 40.7 | 1.1    | 6.7  | 18.3 |
| CE     | 3.1  | 99.7 | 44.2  | 40.7     | —    | 41.9   | 47.5 | 59.1 |
| $GM_1$ | 38.7 | 57.7 | 2.2   | 1.1      | 41.9 | —      | 5.5  | 17.1 |
| CC     | 44.3 | 52.0 | 3.4   | 6.7      | 47.5 | 5.5    | —    | 11.4 |
| SBA    | 55.9 | 40.4 | 15.0  | 18.3     | 59.1 | 17.1   | 11.4 | —    |

threshold, while Balanced Accuracy weighs true positives more heavily, so it prefers a lower threshold. In contrast, $GM_1$, CC, $\kappa$, and $F_1$ agree with each other much better. In Appendix E.1 we conduct a more detailed analysis. In particular, we separately consider the ten-minute and two-hour prediction horizons and show that the behavior and consistency of measures significantly depend on the horizon as the horizon defines the balance between true positives, true negatives, false positives, and false negatives. We also observe that CC and SBA perfectly agree for the ten-minute horizon but have noticeable disagreement for two hours.

## 5.2 Multiclass measures

In this section, we analyze multiclass measures. For all measures that are defined for the multiclass problems, we consider their standard expressions (if not stated otherwise). For other measures ($F_1$, Jaccard, $GM_1$), we use macro averaging.

**Image classification**  We conduct an experiment on ImageNet [24], a classic dataset for image classification. For this, we take the top-10 algorithms that are considered to be state-of-the-art at the moment of submission.[6] We check whether the leaderboard based on accuracy is consistent with the leaderboards based on other measures. Thus, we apply the models to the test set, compute the confusion matrices, and compare all measures defined in Table 1.

Notably, the ImageNet dataset is balanced. This makes all measures more similar to each other. For instance, accuracy and BA are equal in this scenario. Also, the *constant baseline* property discussed in Section 3.5 is especially important for *unbalanced* datasets. Thus, measures are *more consistent* on balanced data. Nevertheless, we notice that the ranking can be inconsistent starting from the algorithm ranked fifth on the leaderboard.

The (partial) results are shown in Table 5. Here we compare EfficientNet-B7 NoisyStudent [31] and Swin-B Transformer (patch size 4x4, window size 12x12, image size $384^2$) [19] that are the fifth and sixth models in the leaderboard. One can see that the measures inconsistently rank the algorithms: Confusion Entropy, Jaccard, and SBA disagree with accuracy and other measures. Interestingly, while Jaccard and $F_1$ always agree for binary problems, they may disagree after the macro averaging, as we see in this case. Also, for one measure, different multiclass extensions may be inconsistent, as we see with macro averaging versus the standard definition of the multiclass Correlation Coefficient. More detailed results can be found in Appendix E.2.

**Sentiment analysis**  In the previous experiment, we noticed that despite several disagreements, the measures usually rank the algorithms similarly. This can be caused by the fact that the test set of ImageNet is balanced: all classes have equal sizes. However, in practical applications, we

---

[6]https://github.com/rwightman/pytorch-image-models/blob/master/results/results-imagenet.csv (May 8, 2021).

Table 5: Inconsistent results on ImageNet, % (fifth and sixth models in the leaderboard)

|  | Acc/BA | $F_1$ | J | $\kappa$ | $1-$CE | $GM_1$ | CC | $CC^{macro}$ | SBA |
|---|---|---|---|---|---|---|---|---|---|
| Efficientnet | **86.46** | **86.30** | 77.525 | **86.44** | 93.41 | **86.28** | **86.44** | 86.419 | 86.57 |
| Swin | 86.43 | 86.27 | **77.531** | 86.42 | **93.51** | 86.26 | 86.42 | **86.423** | **86.61** |

Table 6: Ranking algorithms according to different measures on SST-5: from 1 (best) to 7 (worst)

|  | Acc | BA | $F_1$ | J | $\kappa$ | CE | $GM_1$ | CC | $CC^{macro}$ | SBA |
|---|---|---|---|---|---|---|---|---|---|---|
| Flair+ELMo | 1 | 1 | 1 | 1 | 1 | 1 | 1 | 1 | 1 | 1 |
| Flair+BERT | 2 | 4 | 5 | 5 | 4 | 2 | 5 | 2 | 2 | 2 |
| SVM | 3 | 3 | 3 | 3 | 3 | 5 | 3 | 3 | 4 | 4 |
| Logistic | 4 | 5 | 4 | 4 | 5 | 3 | 4 | 5 | 5 | 3 |
| FastText | 5 | 2 | 2 | 2 | 2 | 6 | 2 | 4 | 3 | 5 |
| VADER | 6 | 6 | 6 | 6 | 6 | 7 | 6 | 6 | 6 | 7 |
| TextBlob | 7 | 7 | 7 | 7 | 7 | 4 | 7 | 7 | 7 | 6 |

rarely encounter balanced data. Thus, we also consider an unbalanced classification task. In this experiment, we take the 5-class Stanford Sentiment Treebank (SST-5) dataset [27]. We compare the following algorithms: TextBlob, VADER, Logistic Regression, SVM, FastText, Flair+ELMo, and Flair+BERT [23]. Table 6 shows that different measures rank the algorithms differently. Among the measures shown in the table, the only consistent rankings are the one provided by $\kappa$ and BA and the second given by $F_1$, GM, and Jaccard. Note that the latter ranking significantly disagrees with the ranking by accuracy.

Appendix E.2 contains an additional experiment with an unbalanced multiclass dataset, where we show the inconsistency rates of the considered measures and different multiclass extensions.

# 6   Conclusion and future work

In this paper, we propose a systematic approach to the analysis of classification performance measures: we propose several desirable properties and theoretically check each property for a list of measures. We also prove an impossibility theorem: some desirable properties cannot be simultaneously satisfied, so either distance or *exact* constant baseline has to be discarded.

Based on the properties we analyzed in this paper, we come to the following practical suggestions. If the distance requirement is needed, Correlation Distance seems to be the best option: it satisfies all the properties except for the exact constant baseline, which is still approximately satisfied. Otherwise, we suggest using one of Generalized Means, including Correlation Coefficient and Symmetric Balanced Accuracy — they satisfy all the properties except distance. For binary classification, CC is a natural choice as it can be non-linearly transformed to a distance. For multiclass problems, Symmetric Balanced Accuracy has an additional advantage: among the considered measures, only this one preserves its good properties in the multiclass case. Finally, we do not advise using averagings, but if needed, then macro averaging preserves more properties.

There are still many open questions and promising directions for future research. First, we would like to see whether one could construct a set of desirable properties that can be used as axioms to uniquely define one good measure (or a parametrized group of measures). Secondly, it is an open problem whether Generalized Means measures in general (or SBA in particular) can be converted to a distance via a continuous transformation. Finally, our work does not cover ranking and probability-based measures. Thus, we leave aside such widely used measures as cross-entropy and AUC. Formalizing and analyzing their properties is an important direction for future research.

**Broader impact**   Our work may help towards reducing certain biases in research. For instance, some measures (e.g., accuracy) are biased towards the majority class. Thus, the bias towards the majority class could be even amplified with the poor metric selection. Our work could provide some clues on how to avoid such a situation.

## Acknowledgments and Disclosure of Funding

Part of this work was done while Martijn Gösgens was visiting Yandex and Moscow Institute of Physics and Technology (Russia). The work of Martijn Gösgens is supported by the Netherlands Organisation for Scientific Research (NWO) through the Gravitation NETWORKS grant no. 024.002.003.

The authors would like to thank Alexander Ganshin, Pert Vytovtov, and Eugenia Elistratova for providing the weather forecasting data.

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
