# A    Related work

Properly choosing an evaluation measure is a significant problem that attracted much attention in recent and long-standing research. In this section, we cover some related papers. In summary, while there are many related studies, the field lacks systematic approaches. Some papers focus on particular advantages and flaws of particular measures, while others suggest some informal properties. Our paper suggests a unified analysis that generalizes and extends the existing research.

A work conceptually related to ours is [26]. In this paper, the authors define a list of properties (they refer to them as *axioms*). Some properties are similar to ours: MON is our monotonicity, FIX is somewhat similar (but not the same) to our maximal and minimal agreement, CHA is the constant baseline, and SYM is our class-symmetry. The properties CON and SDE/WDE are related to singularities. In the current paper, we do not focus on singularities since measures are naturally extended to such cases, as we discuss in Section B. Another property is called Robustness to Imbalance (IMB). This property requires a constant classifier that classifies all elements to either the positive or the negative class to get a constant similarity score $k_1$ or $k_2$, respectively. One can see that our constant baseline thus implies IMB with $k_1 = k_2$. On the other hand, having $k_1 \neq k_2$ may lead to bias towards a particular class, which does not seem to be desired. The authors show that several known measures do not satisfy some of the properties and propose *K measure*, which is a shifted version of Balanced Accuracy with singularities properly resolved. Let us also note that the authors advocate against CC largely because they do not use this same straightforward resolution to the singularities for this measure. Our work differs in the following aspects. First, we consider more comprehensive lists of measures and properties and check each property for each popular measure. In particular, our properties include symmetry (in terms of interchanging labelings), distance, and approximate constant baseline. We show that in terms of the extended list of properties, there are better variants than the K measure (which we refer to as Balanced Accuracy). We also provide a deep theoretical analysis of properties and propose a new family of 'good' measures. In addition, we rigorously analyze the multiclass scenario, including the properties of aggregation schemes. To sum up, while there are methodological similarities, there are significant differences in the analysis and outcomes.

With some similarities to our research, the authors of [13] formulate a list of (informal) properties that are argued to be desirable for an evaluation measure. These properties include having a natural extension to the multiclass case, low complexity and computational cost, distinctiveness and discriminability, informativeness, and favoring the minority class. While informativeness seems to be an informal analog of our *constant baseline*, the properties are not formally defined, and thus no systematic analysis of measures with respect to the properties can be given.

Another work related to our research [28] defines a list of properties by describing several transformations of the confusion matrix that should not change the measure value. As a result, the authors provide a table listing which measures are invariant under which transformations. This analysis includes our *symmetry* and also *scale invariance* which we discuss further in Appendix D. However, the discussed properties are quite simple, and the work does not cover the most important and complex ones like *constant baseline*, *monotonicity*, or *distance*.

There are papers focusing on properties of a particular measure, for instance, Cohen's Kappa [8, 22], Confusion Entropy [7], or Balanced Accuracy [2]. Some papers go beyond the threshold measures considered in our paper. For instance, [6] theoretically analyzes how the area under the ROC curve (AUC) relates to accuracy. Another work focusing on AUC and accuracy is [14]. This paper formally defines two properties: *degree of consistency* and *degree of discriminancy*. The degree of consistency is not a property of a measure but rather a property of a *pair* of measures. In our experiments on synthetic and real data, we compute such degrees of (in)consistency. The degree of discriminancy, in turn, can be reformulated as the number of different values that a measure has (in a given domain).

There are studies advocating using the Matthews correlation instead of some other popular measures. For instance, the authors of [8] compare CC to Cohen's Kappa and show that the latter may have undesirable behavior in some scenarios. Essentially, these scenarios show that Cohen's Kappa does not satisfy our strong monotonicity requirement. A recent paper [3] advocates using CC over $F_1$ and accuracy based on several intuitive *use cases*, where it is clear that the performance is poor, but only CC can correctly detect that in all cases. We note that all the use cases are related to our *constant baseline* property. Similarly to the above research, we conclude that CC should be preferred over $F_1$,

Table 7: Notation

| Variable | Definition |
|---|---|
| n | number of elements |
| m | number of classes |
| $c_{ij}$ | number of elements of class $i$ that are predicted as $j$ |
| $A_i$ | elements with true label $i$ |
| $B_i$ | elements with predicted label $i$ |
| $\mathcal{C} = (c_{ij})$ | $m \times m$ confusion matrix |
| $a_i = \sum_{j=0}^{m-1} c_{ij}$ | size of $i$-th class in the true labeling |
| $b_i = \sum_{j=0}^{m-1} c_{ji}$ | size of $i$-th class in the predicted labeling |
| $p_A = \frac{a_1}{n}$, $p_B = \frac{b_1}{n}$ | fraction of positive entries (for binary classification) |
| $p_{AB} = \frac{c_{11}}{n}$ | fraction of agreeing positives (for binary classification) |
| $M(\mathcal{C})$, $M(A, B)$, $M(p_{AB}, p_A, p_B)$ | classification validation measure |

accuracy, and Cohen's kappa. Importantly, our conclusion is based on a rigorous analysis and formal properties.

Numerous studies empirically compare different classification measures [4, 10]; some of them specifically focus on imbalanced data [20]. Going beyond particular measures, some studies compare the properties of micro- and macro- averagings [29]. However, to the best of our knowledge, our work is the first one giving a formal approach to the problem.

Finally, as we discuss in the main text in more detail, our work is motivated by a recent study [12] that analyzes properties of *cluster validation* measures. We refer to this paper for an overview of related work in cluster analysis.

## B   More on classification validation measures

**Notation**   For convenience, Table 7 lists notation frequently used throughout the text.

**Resolving singularities**   When some of the classes are not present in the predicted (or, more rarely, true) labelings, some measures from Table 1 may not be defined. Let us discuss how to resolve such singularities appropriately.

For some measures, singularities can only occur when the measures maximally or minimally agree with each other. For example, the denominator of Jaccard is only zero if $a_1 = b_1 = 0$, in which case $A = B$ must hold so that the singularity is easily resolved by maximal agreement, leading to $J(A, B) = 1$.

For measures such as Matthews Coefficient, singularities can be resolved using constant baseline. For CC, a singularity can only occur whenever either $n^2 = \sum_{m=1}^{n} a_i^2$ or $n^2 = \sum_{m=1}^{n} b_i^2$. This implies that either $A$ or $B$ classifies all elements to the same class. If both $A$ and $B$ classify all elements to the same class, then the singularity can be resolved by maximal agreement (if they classify to the same class) or minimal agreement (otherwise). If one of $A$ and $B$ classifies all elements to the same class, then the constant baseline tells us that $M(A, B) = 0$ should hold.

Similarly, some measures, e.g., BA and SBA, contain terms $c_{ii}/a_i$ (or $c_{ii}/b_i$) that may have singularities. In cases where $a_i = 0$, these singularities can be algebraically resolved by $c_{ii} = 0 = \frac{a_i b_i}{n}$. This leads to $\frac{c_{ii}}{a_i} = \frac{b_i}{n}$ and ensures that such singularities will not lead to violations of constant baseline.

**Correspondence with pair-counting cluster validation measures**   As discussed in the main text, there is a correspondence between pair-counting cluster validation measures and binary classification validation measures. We refer to Table 8 for some corresponding pairs.

Table 8: Correspondence of binary classification measures and pair-counting clustering measures

| Classification | Clustering |
| --- | --- |
| $F_1$ | Dice |
| Jaccard | Jaccard |
| Matthews Correlation Coefficient | Pearson Correlation Coefficient |
| Accuracy | Rand |
| Cohen's Kappa | Adjusted Rand |
| Symmetric Balanced Accuracy | Sokal&Sneath |
| Correlation Distance | Correlation Distance |

## C   Checking the properties

Table 2 lists which measures satisfy the discussed properties and which averaging schemes preserve them. In this section, we formally prove all the results. Recall that if a measure does not have a natural extension to the multiclass case, then we analyze its binary variant. Additionally, if a property is violated in the binary case, then we do not check it in the multiclass case.

**Using existing analysis of cluster validation indices**   As discussed in the previous section, there is a correspondence between some pair-counting clustering evaluation measures and classification ones. Recall that a pair-counting clustering measure is a function of $N_{11}$, $N_{10}$, $N_{01}$, and $N_{00}$, where $N_{11}$ is the number of element-pairs belonging to the same cluster in both partitions, $N_{00}$ is the number of pairs belonging to different clusters in both partitions, $N_{10}$ is the number of pairs belonging to the same cluster in the true partition but to different clusters in the predicted partition, and $N_{01}$ is the number of pairs belonging to different clusters in the true partition but to the same cluster in the predicted partition. Thus, pair-counting clustering measures are functions of TP, TN, FP, and FN defined for *classifying element-pairs* into "intra-cluster" and "inter-cluster" pairs. So, replacing $N_{ij}$ by $c_{ij}$ we naturally get a binary classification measure. Some classification evaluation indices have been theoretically analyzed in [12]. Using Table 8, we can adopt some of these results for classification measures.

### C.1   Maximal and minimal agreement

To check whether a measure has the maximal or minimal agreement properties, we substitute the entries of a diagonal matrix or a matrix with zero diagonal into the expression: we need either a strict upper or a strict lower bound for the measure values. Note that for measures having the monotonicity property (i.e., for all considered measures except CE and multiclass $\kappa$, CC, CD), it is sufficient to check that we obtain constant values for diagonal and non-diagonal matrices. Indeed, each confusion matrix can be monotonically transformed to a diagonal (or a zero-diagonal) one.

By substituting a diagonal confusion matrix, we get the maximal agreement for $F_1$, J, CC, Acc, BA, $\kappa$, SBA, and $GM_r$ with $c_{\max} = 1$. For $-CD$, the maximal agreement holds with $c_{\max} = 0$. Finally, $CE = 0$ if $\mathcal{C}$ is diagonal and otherwise there exists a pair $(i, j)$ such that $c_{ij} > 0, a_i > 0, b_j > 0$, so we get $-CE < 0$.

The minimal agreement for accuracy, Balanced Accuracy, and Symmetric Balanced Accuracy clearly holds with $c_{\min} = 0$. Substituting a zero-diagonal confusion matrix into $GM_r$, we get $c_{\min} = -1$.

For binary measures $F_1$ and Jaccard, the minimal agreement does not hold: these measures equal zero not only for zero-diagonal matrices, but also when $c_{11} = 0$ and $c_{00} > 0$.

In the binary case, the minimal agreement of CC is satisfied with $c_{\min} = -1$. However, this property is violated if $m > 2$. For instance, consider the confusion matrices $\mathcal{C}_1 = \begin{pmatrix} 0 & 1 & 0 \\ 0 & 0 & 1 \\ 2 & 0 & 0 \end{pmatrix}$ and $\mathcal{C}_2 = \begin{pmatrix} 0 & 1 & 0 \\ 1 & 0 & 1 \\ 0 & 1 & 0 \end{pmatrix}$. We have $CC(\mathcal{C}_1) \neq CC(\mathcal{C}_2)$ (-0.5 and -0.6, respectively), while $\mathcal{C}_1$ and $\mathcal{C}_2$ are both zero-diagonal. Note that CD is a monotone transformation of CC, so CD inherits the same properties.

For CE, the minimal agreement does not hold even in the binary case [7]: let $C_1 = \left(\begin{smallmatrix} 0 & 6 \\ 6 & 0 \end{smallmatrix}\right)$ and $C_2 = \left(\begin{smallmatrix} 1 & 5 \\ 5 & 1 \end{smallmatrix}\right)$. Then, we have $\mathrm{CE}(C_1) = 1$ and $\mathrm{CE}(C_2) > 1$. This contradicts both the minimal agreement and monotonicity properties.

Finally, substituting a zero-diagonal matrix into Cohen's Kappa, we get $\frac{-\sum_i a_i b_i}{n^2 - \sum_i a_i b_i}$ which is clearly non-constant.

## C.2 Symmetry

**Class-symmetry** Almost all considered measures are class-symmetric: they do not change after interchanging class labels. The only exceptions are $F_1$ and Jaccard. Class-symmetry of GM follows from the fact that it can be rewritten as $(c_{11}c_{00} - c_{01}c_{10}) / \left( \sqrt[r]{\frac{1}{2}\left(a_1^r a_0^r + b_1^r b_0^r\right)} \right)$.

**Symmetry** This property is easily verified by swapping $a_i$ with $b_i$ and $c_{ij}$ with $c_{ji}$. Thus, all measures except BA are symmetric.

## C.3 Distance

We refer to [16] for the proof that Jaccard satisfies this requirement. To show that accuracy has this property, we need to show that $1 - \mathrm{Acc}$ is a distance, which is true since $n(1 - \mathrm{Acc})$ is the Hamming distance.

Now, we need to prove that CD is a distance since it was previously known only for the binary case.

**Lemma 1.** *The Correlation Distance* $\mathrm{CD} = \frac{1}{\pi}\arccos(\mathrm{CC})$ *is a distance for any* $m \geq 2$.

*Proof.* Let us represent a classification by a matrix via one-hot encoding, i.e., $A = (a_{ij})_{i\in[n], j\in[m]}$, where $a_{ij} = \mathbb{1}\{A(i) = j\}$, and define $a_j = \sum_i a_{ij}$. Note that for two labelings $A$ and $B$, the Frobenius inner product is given by

$$\langle A, B\rangle = \sum_j c_{jj},$$

where $c_{jj}$ is the $j$-th diagonal entry of the confusion matrix for $A$ and $B$. Next, we define

$$\bar{A} := \left(a_{ij} - \frac{a_j}{n}\right)_{i\in[n], j\in[m]}.$$

Then, for two labelings $A$ and $B$, the Frobenius inner product of these mappings is given by

$$\langle \bar{A}, \bar{B}\rangle = \sum_j \left(c_{jj} - \frac{a_j b_j}{n}\right).$$

And the squared length equals

$$\|\bar{A}\|^2 = n - \frac{\sum_j a_j^2}{n}.$$

Therefore, we get

$$\mathrm{CC}(\mathcal{C}) = \frac{\langle \bar{A}, \bar{B}\rangle}{\|\bar{A}\| \cdot \|\bar{B}\|},$$

so that its arccosine is indeed the angle between $\bar{A}$ and $\bar{B}$, which is a metric distance. $\qquad\square$

Let us now prove that the remaining measures cannot be linearly transformed to metric distances. According to Theorem 1, a measure that satisfies monotonicity and constant baseline cannot have the distance property. This proves that CC, BA, $\kappa$, SBA, and $\mathrm{GM}_r$ cannot be linearly transformed to a distance (note that BA is also not symmetric). To show that CE does not have this property, we take $A = (1, 1, 0)$, $B = (1, 1, 1)$, $C = (1, 0, 1)$. Note that $\mathrm{CE}(A, C) = 1$ and $\mathrm{CE}(A, B) = \mathrm{CE}(B, C) \approx 0.387$. Hence, $\mathrm{CE}(A, C) > \mathrm{CE}(A, B) + \mathrm{CE}(B, C)$ that disproves the distance property. Finally, the counter-example for $F_1$ is given in [12] since $F_1$ is equivalent to the Dice index.

## C.4 Monotonicity

**Strong monotonicity**   $F_1$ and Jaccard are constant w.r.t. $c_{00}$, so they are not strongly monotone. Cohen's Kappa also violates this property [12]: we have $\kappa \left( \begin{smallmatrix} 1 & 2 \\ 1 & 0 \end{smallmatrix} \right) < \kappa \left( \begin{smallmatrix} 1 & 3 \\ 1 & 0 \end{smallmatrix} \right)$. Then, CE is not strongly monotone since it is not monotone (see below).

The fact that CC is strongly monotone in the binary case is proven in [12] (for general binary vectors). In contrast to the binary case, CC is not strongly monotone if $m \geq 3$ since it is not monotone. CD inherits monotonicity properties from CC.

To prove that accuracy is strongly monotone, we use the inequality $(a + x)/(b + x) > a/b$ for $b > a > 0$ and $x > 0$. So, accuracy increases if we simultaneously increment $c_{ii}$ (for some $i$) and $n$. If we increment $n$ and $c_{ij}$ for $i \neq j$, then accuracy decreases, which proves strong monotonicity. Similar reasoning works for BA and SBA.

Finally, let us prove that $\text{GM}_r$ is strongly monotone for any $r$.

**Lemma 2.** $\text{GM}_r$ *is strongly monotone.*

*Proof.* Note that $r \to 0$ corresponds to CC. Since this measure is considered above, we may assume that $r \neq 0$.

Due to the symmetry of GM, we only need to prove that the measure is strongly monotone w.r.t. $c_{11}$ and $c_{10}$. Moreover, GM flips the sign if we invert the labels in one classification. Hence, we only need to prove that it is increasing in $c_{11}$. Considering GM as a function of independent variables $c_{11}, c_{00}, c_{01}, c_{10}$, we calculate

$$\frac{\partial \text{GM}_r}{\partial c_{11}} = (n + c_{11} - b_1 - a_1) \left( \frac{1}{2} \left( a_1^r a_0^r + b_1^r b_0^r \right) \right)^{-1/r}$$

$$- \frac{1}{2r} \left( a_1^{r-1} a_0^r r + b_1^{r-1} b_0^r r \right) (nc_{11} - a_1 b_1) \left( \frac{1}{2} \left( a_1^r a_0^r + b_1^r b_0^r \right) \right)^{-1/r-1}.$$

Simplifying the expression, we note that it has the same sign as the following sum

$$(n + c_{11} - b_1 - a_1) \left( a_1^r a_0^r + b_1^r b_0^r \right) - \left( a_1^{r-1} a_0^r + b_1^{r-1} b_0^r \right) (nc_{11} - a_1 b_1)$$

$$= a_0^r a_1^{r-1} \left( -nc_{11} + a_1 b_1 + a_1 n + a_1 c_{11} - b_1 a_1 - a_1^2 \right)$$

$$+ b_0^r b_1^{r-1} \left( -nc_{11} + a_1 b_1 + b_1 n + b_1 c_{11} - b_1 a_1 - b_1^2 \right)$$

$$= a_0^r a_1^{r-1} \cdot a_0 c_{10} + b_0^r b_1^{r-1} \cdot b_0 c_{01} \geq 0.$$

Note that the last expression is strictly positive if the classifications $A$ and $B$ do not coincide and are not constant. $\qquad \square$

**Monotonicity**   First, we note that monotonicity of Acc, BA, SBA, and GM follows from their strong monotonicity. Monotonicity of $F_1$ and Jaccard follows from their definitions, see also [12].

Monotonicity of CC in the binary case follows from its strong monotonicity. However, for $m \geq 3$, CC is not monotone. Indeed, consider $\mathcal{C}_1 = \left( \begin{smallmatrix} 1 & 0 & 0 \\ 6 & 1 & 0 \\ 0 & 0 & 1 \end{smallmatrix} \right), \mathcal{C}_2 = \left( \begin{smallmatrix} 1 & 0 & 0 \\ 7 & 0 & 0 \\ 0 & 0 & 1 \end{smallmatrix} \right)$ and note that $\text{CC}(\mathcal{C}_2) > \text{CC}(\mathcal{C}_1)$.

The fact that Cohen's Kappa is monotone follows from [12] (the proof for Adjusted Rand applies to general binary vectors). Similarly to CC, for $m \geq 3$, monotonicity is violated. Consider, for example, $\mathcal{C}_1 = \left( \begin{smallmatrix} 0 & 1 & 2 \\ 0 & 0 & 0 \\ 1 & 0 & 0 \end{smallmatrix} \right), \mathcal{C}_2 = \left( \begin{smallmatrix} 1 & 0 & 2 \\ 0 & 0 & 0 \\ 1 & 0 & 0 \end{smallmatrix} \right)$ and note that $\kappa(\mathcal{C}_1) > \kappa(\mathcal{C}_2)$.

Finally, the example from Section C.1 disproves the monotonicity of CE.

## C.5 Constant baseline

**Approximate constant baseline**   Substituting $c_{ij} = a_i b_j / n$ into CC, CD, BA, $\kappa$, SBA, and GM, we get values that do not depend on $a_i, b_i$. Thus, these measures have the approximate constant baseline property.

Substituting $c_{ij} = a_i b_j / n$ into CE, we get that the result depends on $a_i$ and $b_j$. For instance, taking $(a_0, a_1) = (2, 1), (b_0, b_1) = (1, 2)$ and $(a_0, a_1) = (0, 3), (b_0, b_1) = (1, 2)$ we get different values of

CE that disproves approximate constant baseline. Similarly, $F_1$, Jaccard, and accuracy do not have this property.

**Exact constant baseline**    We will use the following lemma.

**Lemma 3.** *Suppose that the fixed true labeling $A$ has class-sizes $a_1, \ldots, a_m$, while the predicted labeling $B \sim U(b_1, \ldots, b_m)$ is random. Then, $\mathbb{E}_{B \sim U(b_1, \ldots, b_m)} c_{ij} = a_i b_j / n$.*

*Proof.* To prove this equality, we simply note that

$$\mathbb{E}_{B \sim U(b_1, \ldots, b_m)} c_{ij} = \sum_{x \in A_i} \mathbb{E} \, \mathbb{1}\{x \in B_j\} = a_i \, \mathbb{P}\left(\tilde{x} \in B_j\right) = a_i \, \mathbb{E} \sum_{y \in B_j} \mathbb{1}\{\tilde{x} = y\} = \frac{a_i b_j}{n},$$

where $\tilde{x}$ is an arbitrary element of $A_i$. $\qquad\square$

Now, let us prove that all measures that have the exact constant baseline property also have the approximate constant baseline.

**Lemma 4.** *If a measure $M(\mathcal{C})$ is scale-invariant (see Definition 11), continuous, and has the constant baseline property, then it also has the approximate constant baseline.*

*Proof.* Let us fix non-negative numbers $\{a_i\}_{i=0}^{m-1}, \{b_i\}_{i=0}^{m-1}$ such that $\sum_{i=0}^{m-1} a_i = \sum_{i=0}^{m-1} b_i = n$. Then, consider a fixed classification $A^N$ with class sizes $Na_1, \ldots, Na_m$ and a random classification $B^N$ taken from $U(Nb_1, \ldots, Nb_m)$.

Let $c_{ij}^N$ denote entries of the confusion matrix for $A^N$ and $B^N$. Let us prove that for any $i, j \in \{1, \ldots m\}$, the random variable $c_{ij}^N / N$ converges to $a_i b_j / n$ in $L_2$ as $N \to \infty$. From Lemma 3, we have $\mathbb{E}(c_{ij}/N) = a_i b_j / n$. Let us compute $\mathrm{Var}(c_{ij})$. Recall that $c_{ij} = \sum_{x \in A_i^N} \mathbb{1}\{x \in B_j^N\}$, then

$$\mathrm{Var}(c_{ij}) = \sum_{x, y \in A_i^N} \mathrm{Cov}\left(\mathbb{1}\{x \in B_j^N\}, \mathbb{1}\{y \in B_j^N\}\right).$$

It remains to compute $\mathrm{Cov}\left(\mathbb{1}\{x \in B_j^N\}, \mathbb{1}\{y \in B_j^N\}\right)$ for $x = y$ and $x \neq y$. For this, note that

$$\mathbb{P}\left(x \in B_j^N\right) = b_j / n \text{ and } \mathbb{P}\left(x, y \in B_j^N\right) = \frac{Nb_j(Nb_j - 1)}{Nn(Nn - 1)} \text{ for } x \neq y.$$

Then,

$$\mathrm{Cov}\left(\mathbb{1}\{x \in B_j^N\}, \mathbb{1}\{y \in B_j^N\}\right) = \mathbb{P}\left(x, y \in B_j^N\right) - \left(\mathbb{P}\left(x \in B_j^N\right)\right)^2 = O(1/N).$$

Thus, we get that $\mathrm{Var}(c_{ij}/N) = O(N)/N^2 = O(1/N)$ and prove $L_2$-convergence.

Now we are ready to prove the lemma. Let $M$ be a scale-invariant, continuous measure that has constant baseline. Then,

$$c_{\text{base}} = \mathbb{E}_{B^N \sim U(Nb_1, \ldots, Nb_m)} M(\mathcal{C}_N) = \mathbb{E} M\left(\frac{\mathcal{C}^N}{N}\right) \xrightarrow[N \to \infty]{} M(\mathcal{C}),$$

where $\mathcal{C}^N$ is the confusion matrix for $A^N$ and $B^N$ and $\mathcal{C}$ is the confusion matrix for $A$ and $B$. Here $\mathbb{E} M\left(\mathcal{C}^N / N\right) \to M(\mathcal{C})$ holds since the $L_2$-convergence of $c_{ij}^N$ to $a_i b_j / n$ implies convergence in distribution. $\qquad\square$

From this lemma, we get that $F_1$, Jaccard, Acc, and CE do not have constant baseline since they violate the approximate constant baseline property.

Assume that a measure $M(\mathcal{C})$ is linear in $c_{ii}$ for fixed $a_i$ and $b_j$. Then, using the linearity of expectation, we note that approximate constant baseline implies exact constant baseline for such measures. This observation gives that CC, BA, $\kappa$, SBA, and $GM_r$ have the constant baseline property.

Finally, we note that CD violates the constant baseline property as it has both monotonicity and distance properties (in binary case), while Theorem 1 states that all three properties cannot be simultaneously satisfied.

### C.6 Preserving properties by averagings

**Micro averaging** Recall that for micro averaging, we sum up the binary confusion matrices corresponding to $m$ one-vs-all classifications. Formally, we set $\text{TP} := \sum_{i=0}^{m-1} c_{ii}$, $\text{FN} := \text{FP} = n - \sum_{i=0}^{m-1} c_{ii}$, $\text{TN} := (m-2)n + \sum_{i=0}^{m-1} c_{ii}$. Then, we compute the binary measure.

First, it is easy to see that this averaging preserves symmetry and class-symmetry.

Let us prove that micro averaging preserves the maximal agreement property. If a confusion matrix $\mathcal{C}$ is diagonal, then $n - \sum_{i=0}^{m-1} c_{ii} = 0$ and $\text{FP} = \text{FN} = 0$. Substituting these values in a binary measure $M$, we get $c_{\max}$. If $\mathcal{C}$ is not diagonal, then $\text{FP} = \text{FN} = n - \sum_{i=0}^{m-1} c_{ii} > 0$ and the result of the averaging will be strictly lower than $c_{\max}$. On the other hand, minimal agreement is not preserved since $\text{TN} = (m-2)n > 0$ for zero-diagonal confusion matrices. As a simple example, consider a measure $\mathbb{1}\{\text{TP} + \text{TN} > 0\}$ satisfying the minimal agreement property. Then, after micro averaging, this measure is constant, thus violating minimal agreement.

Also, micro averaging preserves monotonicity: increasing $c_{ii}$ for fixed $n$ leads to increased TP and TN, leaving TP+FP, TP+FN, TN+FP, TN+FN unchanged. On the other hand, strong monotonicity can be violated: incrementing $c_{ij}$ for $i \neq j$ we increase $n$, so $\text{TN} = (m-2)n + \sum_{i=0}^{m-1} c_{ii}$ also increases and the averaged measure may increase. For example, consider a strongly monotone binary measure $\text{TP} + \text{TN} - \text{FP} - \text{FN}$. Then, after micro averaging, it reduces to $nm$, which violates strong monotonicity.

To prove that micro averaging preserves the distance property, we first note that it preserves maximal agreement and symmetry. To show that the triangle inequality is also preserved, we consider micro averaging as a result of the following procedure. First, we use one-hot encoding to map each class to a binary vector. Then, we map a classification vector $A$ of size $n$ to the binary vector $\hat{A}$ of size $nm$ consisting of one-hot encoded binary vectors. Finally, for two classifications $A$ and $B$, we compute the binary measure for $\hat{A}$ and $\hat{B}$. It is easy to see that this procedure is equivalent to micro averaging. Thus, for any multiclass labelings $A, B, C$, there exist binary labelings $\hat{A}, \hat{B}, \hat{C}$ with confusion matrices corresponding to the result of micro averaging. Hence, the triangle inequality for micro averaged matrices follows from the binary property.

Finally, approximate constant baseline can be violated after micro averaging. Indeed, let us take $c_{ii} = a_i b_i / n$. Then, after the averaging, we get $\text{TP} = \sum_{i=0}^{m-1} a_i b_i / n$, which is not necessary equal to $(\text{TP} + \text{FN})(\text{TP} + \text{FP})/(mn) = n/m$. As an example, we can consider a measure $\text{TP} - (\text{TP} + \text{FP})(\text{TP} + \text{FN})/(\text{TP} + \text{FP} + \text{FN} + \text{TN})$ having constant baseline. Thus, the averaged measure is $\sum_{i=0}^{m-1} c_{ii} - n/m$, which does not have an approximate constant baseline. Consequently, the constant baseline property is also violated.

**Macro averaging** As for the micro averaging, symmetry and class-symmetry are clearly satisfied.

Let us check the maximal agreement. Consider a binary measure $M$ having this property. If $\mathcal{C}$ is diagonal, then the result of the averaging is $\frac{1}{m} \sum_i M(c_{ii}, 0, 0, n - c_{ii}) = c_{\max}$. If $\mathcal{C}$ is not diagonal, then one of $a_i - c_{ii} > 0$ and the averaged measure is strictly lower than $c_{\max}$. In contrast, the minimal agreement property can be violated, since for a zero-diagonal confusion matrix the result of the averaging is $\frac{1}{m} \sum_i M(0, a_i, b_i, n - a_i - b_i)$. Since we may have $n - a_i - b_i > 0$, the minimal agreement can be violated. For instance, consider the measure $\mathbb{1}\{\text{TP} + \text{TN} > 0\}$ satisfying the minimal agreement property in the binary case. Then, taking $\mathcal{C}_1 = \left( \begin{smallmatrix} 0 & 1 & 0 \\ 0 & 0 & 1 \\ 1 & 1 & 0 \end{smallmatrix} \right)$ and $\mathcal{C}_2 = \left( \begin{smallmatrix} 0 & 0 & 1 \\ 0 & 0 & 1 \\ 1 & 1 & 0 \end{smallmatrix} \right)$ we get that the averaging has different values on these matrices (1 and $2/3$, respectively), thus the minimal agreement property does not hold.

It is easy to see that monotonicity is preserved by macro averaging. However, strong monotonicity can be violated. Indeed, assume that $c_{ij}$ increases. Then, for $k \notin \{i, j\}$, the values $c_{kk}, a_k, b_k$ do not change while $n$ increases. To show that this can break strong monotonicity, consider the same counterexample as for the micro averaging: $\text{TP} + \text{TN} - \text{FP} - \text{FN}$. Then, after macro averaging, we get the measure $\left( n(m-4) + 4 \sum_{i=0}^{m-1} c_{ii} \right)/m$ that is not strongly monotone.

Let us prove that macro averaging preserves the distance property. As for the micro averaging, it remains to check the triangle inequality. Let $A$, $B$, and $C$ be multiclass classifications with $n$

elements and $m$ classes. Then, for all $i \in \{1, \ldots, m\}$, we can build the binary labelings $A^i, B^i, C^i$ corresponding to one-vs-all binary classifications. Triangle inequality holds for each $A^i, B^i, C^i$. Thus, summing up these inequalities over all $i \in \{1, \ldots, m\}$, we prove the triangle inequality for the macro-averaged measure.

Finally, approximate and exact constant baseline are preserved by the macro averaging due to the linearity of expectation.

**Weighted averaging** Similar reasoning as above, allows one to show that weighted averaging preserves the maximal agreement, class-symmetry, monotonicity, exact and approximate constant baseline.

For the minimal agreement, the counterexample used for macro averaging also works in this case.

Clearly, weighted averaging is not symmetric: we normalize by the class sizes $a_i$. Therefore, the distance property is not preserved as it requires symmetry.

Finally, as a counterexample to strong monotonicity, we can take $M = \text{TP} + \text{TN} - \text{FP} - \text{FN}$ and $\mathcal{C}_1 = \left( \begin{smallmatrix} 0 & 1 & 1 \\ 1 & 0 & 0 \\ 1 & 0 & 0 \end{smallmatrix} \right), \mathcal{C}_2 = \left( \begin{smallmatrix} 0 & 1 & 1 \\ 1 & 0 & 1 \\ 1 & 0 & 0 \end{smallmatrix} \right)$, Then, $M(\mathcal{C}_1) = -2 < -9/5 = \mathcal{C}_2)$.

# D    Theoretical analysis

In this section, we perform a theoretical analysis of binary classification measures. First, we generalize the definition of constant baseline and theoretically compare the two non-linear distance-transformations of the Matthews Correlation Coefficient. Then, we derive the class of measures that satisfy all properties except distance.

## D.1    Higher-order approximate constant baseline

Before we generalize our definition of constant baseline, let us introduce some additional properties. These properties differ from the properties introduced in the main text in the sense that they are not desirable in themselves but are rather *instrumental* for the analysis of other desirable properties.

**Definition 11.** *A measure $M$ is* scale-invariant *if, for any scalar $\alpha > 0$ and confusion matrix $\mathcal{C}$, $M(\alpha\mathcal{C}) = M(\mathcal{C})$.*

We remark that all measures of Table 1 are scale-invariant.

Note that any binary classification measure can be written as a function of the four variables $c_{11}, a_1, b_1, n$ as $c_{10} = a_1 - c_{11}$, $c_{01} = b_1 - c_{11}$, and $c_{00} = n - a_1 - b_1 + c_{11}$. Therefore, any scale-invariant binary classification measure can be written as a function of the three fractions $p_{AB} = c_{11}/n$, $p_A = a_1/n$, and $p_B = b_1/n$. Hence, we will use the shorthand notation $M(\mathcal{C}) = M(p_{AB}, p_A, p_B)$ for the remainder of this analysis. We will write $P_{AB}$ instead of $p_{AB}$ whenever $B$ is random. Note that for $B \sim U(p_B n, (1 - p_B)n)$, it holds that $\mathbb{E}_{B \sim U(p_B n, (1-p_B)n)}[P_{AB}] = p_A p_B$. Thus, it can readily be seen that the approximate constant baseline is satisfied whenever $M(p_A p_B, p_A, p_B) = c_{\text{base}}$. We introduce one additional property that ensures that the measure is a well-behaved function in terms of these variables.

**Definition 12.** *A scale-invariant measure $M$ is* smooth *if, for any $p_A, p_B \in (0, 1)$, the Taylor series of $M(p_{AB}, p_A, p_B)$ around the point $p_{AB} = p_A p_B$ converges absolutely on the interval $p_{AB} \in [0, \min\{p_A, p_B\}]$. That is, for all $p_A, p_B \in (0, 1)$ and $p_{AB} \in [0, \min\{p_A, p_B\}]$, we have*

$$\sum_{k=0}^{\infty} \left| \frac{(p_{AB} - p_A p_B)^k}{k!} \frac{\partial^k}{\partial p_{AB}^k} M(p_A p_B, p_A, p_B) \right| < \infty.$$

Note that such absolute convergence implies that the Taylor series converges to $M(p_{AB}, p_A, p_B)$. We remark that all constant-baseline measures of Table 1 are linear functions in $p_{AB}$ for fixed $p_A, p_B$. Thus, each of these is smooth. Furthermore, because CC is linear in $p_{AB}$, we have that for any transformation $f(\text{CC})$, the Taylor expansion of $f(\text{CC})$ is given by substituting CC in the Taylor expansion of $f$. Thus, since the Taylor expansion of $f_1(x) = \frac{1}{\pi} \arccos(x)$ and $f_2(x) = \sqrt{2(1 - x)}$ around $x = 0$ converges for $x \in [-1, 1]$, we have that CD$= f_1(\text{CC})$ and CD$' = f_2(\text{CC})$ are also smooth measures.

This allows us to express the expected value of a measure in terms of the central moments of $P_{AB}$:

$$\mathbb{E}[M(P_{AB}, p_A, p_B)] = \mathbb{E}\left[\sum_{k=0}^{\infty} \frac{(P_{AB} - p_A p_B)^k}{k!} \frac{\partial^k}{\partial p_{AB}^k} M(p_A p_B, p_A, p_B)\right]$$

$$= \sum_{k=0}^{\infty} \frac{\mathbb{E}[(P_{AB} - p_A p_B)^k]}{k!} \frac{\partial^k}{\partial p_{AB}^k} M(p_A p_B, p_A, p_B).$$

Here, the absolute convergence helps bound the term inside the expectation so that the Dominated Convergence Theorem allows us to interchange summation and expectation. In this expression, the first-order term vanishes as $\mathbb{E}[P_{AB}] = p_A p_B$. Thus, we have

$$\mathbb{E}[M(P_{AB}, p_A, p_B)] = M(p_A p_B, p_A, p_B) + \sum_{k=2}^{\infty} \frac{\mathbb{E}[(P_{AB} - p_A p_B)^k]}{k!} \frac{\partial^k}{\partial p_{AB}^k} M(p_A p_B, p_A, p_B).$$

Note that for large numbers of items, $P_{AB}$ is highly concentrated around $p_A p_B$. Thus, the contribution of the higher-order central moments is relatively small. This leads to the following generalization of the constant baseline.

**Definition 13.** *A smooth measure $M$ has a $k$-th order approximate constant baseline, if there exists a constant $c_{base}$ such that $M(p_A p_B, p_A, p_B) = c_{base}$, while for all $\ell \in \{2, \ldots, k\}$, it holds that*

$$\frac{\partial^\ell}{\partial p_{AB}^\ell} M(p_A p_B, p_A, p_B) = 0.$$

Thus, first-order constant baseline is equivalent to the approximate constant baseline. Furthermore, note that $\infty$-th order approximate constant baseline implies exact constant baseline since then

$$\mathbb{E}[M(P_{AB}, p_A, p_B)] = M(p_A p_B, p_A, p_B) = c_{\text{base}}.$$

While it seems likely that the exact constant baseline also implies $\infty$-th order constant baseline, we were not able to formally prove this. However, all constant-baseline measures of Table 1 also satisfy $\infty$-th order constant baseline. For this reason, we will use $\infty$-th order constant baseline as a substitute for the exact constant baseline when deriving measures from properties.

### D.2 Constant baseline order of distance transformations

We now show that the constant baseline of $\text{CD} = \frac{1}{\pi} \arccos(\text{CC})$ is one order higher than $\text{CD}' = \sqrt{2(1 - \text{CC})}$.

**Statement 6.** $\text{CD} = \frac{1}{\pi} \arccos(\text{CC})$ *has a second-order approximate constant baseline while $\text{CD}' = \sqrt{2(1 - \text{CC})}$ only has a first-order approximate constant baseline.*

*Proof.* The Matthews Correlation Coefficient is given by

$$\text{CC}(p_{AB}, p_A, p_B) = \frac{p_{AB} - p_A p_B}{\sqrt{p_A(1 - p_A)p_B(1 - p_B)}},$$

so that it is indeed a linear function in $p_{AB}$ for fixed $p_A, p_B$. Therefore, the Taylor expansions of CD and CD$'$ are obtained by simply substituting CC into the Taylor expansions of $\frac{1}{\pi} \arccos(x)$ and $\sqrt{2(1 - x)}$ respectively. We have

$$\frac{1}{\pi} \arccos(x) = \frac{\pi}{2} - \sum_{k=0}^{\infty} \frac{(2k)! x^{2k+1}}{4^k (k!)^2 (2k+1)} \text{ and } \sqrt{2(1 - x)} = \sqrt{2} - \sqrt{2} \sum_{k=0}^{\infty} \frac{2}{k+1} \binom{2k}{k} \left(\frac{x}{4}\right)^{k+1}.$$

Thus, we see that $\sqrt{2(1 - x)}$ we have a quadratic term, which we do not have for $\frac{1}{\pi} \arccos(x)$. This shows that CD has a second-order constant baseline while CD$'$ only has a first-order constant baseline. $\square$

### D.3 Deriving measures satisfying all properties except distance

Let us derive a class of measures satisfying all properties from Table 2 except distance. We will use $\infty$-th order constant baseline instead of the exact constant baseline as this property is easier to analyze while it implies exact constant baseline and coincides with it for all measures of Table 1.

**Theorem 3.** *Let $M$ be a smooth binary classification measure that satisfies the following properties:*

1. *$\infty$-th order constant baseline with constant $0$;*

2. *symmetry;*

3. *class-symmetry;*

4. *maximal agreement with constant $1$;*

5. *minimal agreement with constant $-1$;*

6. *strong monotonicity.*

*Then, it is of the following form:*

$$M(p_{AB}, p_A, p_B) = s(p_A, p_B)(p_{AB} - p_A p_B),$$

*where $s$ satisfies the following properties:*

1. *$s(p_B, p_A) = s(p_A, p_B) = s(1 - p_A, 1 - p_B)$;*

2. *$s(p_A, p_A) = s(p_A, 1 - p_A) = \frac{1}{p_A(1 - p_A)}$;*

3. *$s(p_A, p_B) < \max\left\{\frac{1}{p_A p_B}, \frac{1}{(1 - p_A)(1 - p_B)}\right\}$ for $p_B \neq 1 - p_A$;*

4. *$s(p_A, p_B) < \max\left\{\frac{1}{p_A(1 - p_B)}, \frac{1}{(1 - p_A)p_B}\right\}$ for $p_B \neq p_A$;*

5. *$\frac{1}{s}\left(p_A \frac{\partial}{\partial p_A} + p_B \frac{\partial}{\partial p_B}\right)s \in \left[\min\left\{-2, -1 - \frac{p_A p_B}{(1 - p_A)(1 - p_B)}\right\}, \max\left\{\frac{2p_B - 1}{1 - p_B}, \frac{2p_A - 1}{1 - p_A}\right\}\right]$;*

6. *$\frac{1}{s}\left((1 - p_A)\frac{\partial}{\partial p_A} - p_B \frac{\partial}{\partial p_B}\right)s \in \left[\min\left\{2 - \frac{1}{p_A}, 2 - \frac{1}{1 - p_B}\right\}, \max\left\{1 + \frac{p_B(1 - p_A)}{p_A(1 - p_B)}, 2\right\}\right]$.*

*Proof.* From the definition of $\infty$-th order constant baseline, we have that $M(p_{AB}, p_A, p_B)$ must be a linear function in $p_{AB}$ for fixed $p_A, p_B$. Thus, it must be of the form

$$M(p_{AB}, p_A, p_B) = c_{\text{base}} + (p_{AB} - p_A p_B)s(p_A, p_B) = (p_{AB} - p_A p_B)s(p_A, p_B)$$

for some function $s(\cdot, \cdot)$.

Now, symmetry requires $M(p_{AB}, p_B, p_A) = M(p_{AB}, p_A, p_B)$ which leads to $s(p_B, p_A) = s(p_A, p_B)$. Then, class-symmetry requires $M(p_{AB}, p_A, p_B) = M(1 - p_A - p_B + p_{AB}, 1 - p_A, 1 - p_B)$, leading to $s(1 - p_A, 1 - p_B) = s(p_A, p_B)$.

For maximal agreement, we have $M(p_{AB}, p_A, p_B) \leq 1$ with equality only if $p_{AB} = p_A = p_B$, i.e., $M(p_A, p_A, p_A) = 1$, leading to $s(p_A, p_A) = \frac{1}{p_A(1 - p_A)}$. Furthermore, $M(p_{AB}, p_A, p_B) \leq M(\min\{p_A, p_B\}, p_A, p_B) < 1$ for $p_A \neq p_B$ is satisfied by

$$s(p_A, p_B) < \frac{1}{\min\{p_A, p_B\} - p_A p_B} = \frac{1}{\min\{p_A(1 - p_B), (1 - p_A)p_B\}}$$

$$= \max\left\{\frac{1}{p_A(1 - p_B)}, \frac{1}{(1 - p_A)p_B}\right\}.$$

Minimal agreement requires $M(p_{AB}, p_A, p_B) \geq -1$ with equality only if $p_{AB} = 0, p_B = 1 - p_A$. For equality, we need

$$s(p_A, 1 - p_A) = \frac{1}{p_A(1 - p_A)}.$$

While for $p_B \neq 1 - p_A$, we need $M(p_{AB}, p_A, p_B) \geq M(\max\{0, p_A + p_B - 1\}, p_A, p_B) > -1$, leading to

$$s(p_A, p_B) < \frac{1}{\min\{p_A p_B, (1 - p_A)(1 - p_B)\}} = \max\left\{\frac{1}{p_A p_B}, \frac{1}{(1 - p_A)(1 - p_B)}\right\}.$$

For the remainder of the proof, we will derive that strong monotonicity is satisfied when the last two conditions of Theorem 3 hold. The first one will be derived from the increasingness of $M$ in $N_{00}$ while the second one will be derived from decreasingness in $N_{10}$. Increasingness in $N_{11}$ and decreasingness in $N_{01}$ will then follow from class-symmetry and symmetry, respectively.

We rewrite the condition $\frac{d}{dN_{00}} M$ to

$$\frac{d}{dN_{00}} M\left(\frac{N_{11}}{N_{11} + N_{10} + N_{01} + N_{00}}, \frac{N_{11} + N_{10}}{N_{11} + N_{10} + N_{01} + N_{00}}, \frac{N_{11} + N_{01}}{N_{11} + N_{10} + N_{01} + N_{00}}\right)$$

$$= -\frac{1}{N}\left[p_{AB}\frac{\partial}{\partial p_{AB}} + p_A\frac{\partial}{\partial p_A} + p_B\frac{\partial}{\partial p_B}\right] M(p_{AB}, p_A, p_B).$$

Since we want $\frac{d}{dN_{00}} M > 0$, we need

$$\left[p_{AB}\frac{\partial}{\partial p_{AB}} + p_A\frac{\partial}{\partial p_A} + p_B\frac{\partial}{\partial p_B}\right] M(p_{AB}, p_A, p_B) < 0.$$

We compute the partial derivatives of $M$:

$$\frac{\partial}{\partial p_{AB}} M = s,$$
$$\frac{\partial}{\partial p_A} M = -p_B s + (p_{AB} - p_A p_B)\frac{\partial}{\partial p_A} s, \tag{3}$$
$$\frac{\partial}{\partial p_B} M = -p_A s + (p_{AB} - p_A p_B)\frac{\partial}{\partial p_B} s.$$

Thus, we need

$$(p_{AB} - 2p_A p_B) \cdot s + (p_{AB} - p_A p_B)\left[p_A\frac{\partial}{\partial p_A} + p_B\frac{\partial}{\partial p_B}\right] s < 0$$

for all $p_{AB} \in [\max\{p_A + p_B - 1, 0\}, \min\{p_A, p_B\}]$. Since the left-hand side is linear in $p_{AB}$, we only need to check the upper and lower limit. Substituting $p_{AB} = \min\{p_A, p_B\}$ leads to

$$\left[p_A\frac{\partial}{\partial p_A} + p_B\frac{\partial}{\partial p_B}\right] s < \frac{2p_A p_B - \min\{p_A, p_B\}}{\min\{p_A, p_B\} - p_A p_B} s$$
$$= \left(\frac{p_A p_B}{\min\{p_A(1 - p_B), p_B(1 - p_A)\}} - 1\right) s$$
$$= \max\left\{\frac{p_B}{1 - p_B} - 1, \frac{p_A}{1 - p_A} - 1\right\} s$$
$$= \max\left\{\frac{2p_B - 1}{1 - p_B}, \frac{2p_A - 1}{1 - p_A}\right\} s.$$

Substituting $p_{AB} = \max\{0, p_A + p_B - 1\}$ gives a lower bound

$$\left[p_A\frac{\partial}{\partial p_A} + p_B\frac{\partial}{\partial p_B}\right] s > -\frac{2p_A p_B - \max\{0, p_A + p_B - 1\}}{p_A p_B - \max\{0, p_A + p_B - 1\}} s$$
$$= -\left(1 + \frac{p_A p_B}{\min\{p_A p_B, (1 - p_A)(1 - p_B)\}}\right) s$$
$$= -\max\left\{2, 1 + \frac{p_A p_B}{(1 - p_A)(1 - p_B)}\right\} s.$$

Combining this, we conclude that increasingness in $N_{00}$ is satisfied whenever it holds that

$$\frac{1}{s}\left(p_A\frac{\partial}{\partial p_A} + p_B\frac{\partial}{\partial p_B}\right) s \in \left[\min\left\{-2, -1 - \frac{p_A p_B}{(1 - p_A)(1 - p_B)}\right\}, \max\left\{\frac{2p_B - 1}{1 - p_B}, \frac{2p_A - 1}{1 - p_A}\right\}\right],$$

as required.

The condition for decreasingness in $N_{10}$ is obtained similarly. The condition $\frac{d}{dN_{10}}M < 0$ can be rewritten to

$$\left[-p_{AB}\frac{\partial}{\partial p_{AB}} + (1-p_A)\frac{\partial}{\partial p_A} - p_B\frac{\partial}{\partial p_B}\right]M(p_{AB}, p_A, p_B) < 0.$$

Substituting the partial derivatives from (3) gives

$$s\cdot(-p_{AB}-(1-p_A)p_B+p_Ap_B)+(p_{AB}-p_Ap_B)\left((1-p_A)\frac{\partial}{\partial p_A}-p_B\frac{\partial}{\partial p_B}\right)s < 0.$$

Again, this linear inequality should hold for all $p_{AB} \in [\max\{p_A+p_B-1,0\}, \min\{p_A,p_B\}]$ and we only need to test the extremes. For $p_{AB} = \min\{p_A, p_B\}$, we find the upper bound

$$\frac{1}{s}\left((1-p_A)\frac{\partial}{\partial p_A}-p_B\frac{\partial}{\partial p_B}\right)s < \frac{\min\{p_A,p_B\}+(1-p_A)p_B-p_Ap_B}{\min\{p_A,p_B\}-p_Ap_B}$$

$$= \frac{\min\{p_A(1-p_B)+p_B(1-p_A), 2p_B(1-p_A)\}}{\min\{p_A(1-p_B), p_B(1-p_A)\}}$$

$$= \max\left\{1+\frac{p_B(1-p_A)}{p_A(1-p_B)}, 2\right\}.$$

Substituting $p_{AB} = \max\{0, p_A+p_B-1\}$, we get the following upper bound

$$\frac{1}{s}\left((1-p_A)\frac{\partial}{\partial p_A}-p_B\frac{\partial}{\partial p_B}\right)s > \frac{\max\{0, p_A+p_B-1\}+(1-p_A)p_B-p_Ap_B}{\max\{0, p_A+p_B-1\}-p_Ap_B}$$

$$= -\frac{\max\{p_B(1-2p_A), p_A+2p_B-1-2p_Ap_B\}}{\min\{p_Ap_B, (1-p_A)(1-p_B)\}}$$

$$= \min\left\{2-\frac{1}{p_A}, 2-\frac{1}{1-p_B}\right\}.$$

Combined, we obtain the desired condition

$$\frac{1}{s}\left((1-p_A)\frac{\partial}{\partial p_A}-p_B\frac{\partial}{\partial p_B}\right)s \in \left[\min\left\{2-\frac{1}{p_A}, 2-\frac{1}{1-p_B}\right\}, \max\left\{1+\frac{p_B(1-p_A)}{p_A(1-p_B)}, 2\right\}\right].$$

$\square$

### D.4 Generalized Means measure

The Generalized Means measure $\mathrm{GM}_r$ corresponds to $s(p_A, p_B) = M_r(p_A(1-p_A), p_B(1-p_B))^{-1}$, where $M_r$ is the generalized mean with exponent $r$.

**Lemma 5.** $s(p_A, p_B) = M_r(p_A(1-p_A), p_B(1-p_B))^{-1}$ *satisfies all the conditions of Theorem 3.*

*Proof.* The proof follows from Section C, where it is shown that $\mathrm{GM}_r$ indeed satisfies all the required properties. Let us also demonstrate the conditions explicitly.

The first four conditions can be easily verified by substituting this $s(p_A, p_B)$. Verifying the last two conditions require a bit more work. The partial derivatives of $s(p_A, p_B)$ are given by

$$\frac{\partial}{\partial p_A}\left[\frac{1}{2}\left(p_A(1-p_A)\right)^r + \frac{1}{2}\left(p_B(1-p_B)\right)^r\right]^{-\frac{1}{r}}$$

$$= -\frac{1}{r}\frac{\frac{r}{2}\left(p_A(1-p_A)\right)^{r-1}(1-2p_A)}{\left[\frac{1}{2}\left(p_A(1-p_A)\right)^r + \frac{1}{2}\left(p_B(1-p_B)\right)^r\right]^{\frac{r+1}{r}}}$$

$$= \frac{2p_A-1}{p_A(1-p_A)}\cdot\frac{\left(p_A(1-p_A)\right)^r}{\left(p_A(1-p_A)\right)^r + \left(p_B(1-p_B)\right)^r}\cdot s,$$

and similarly
$$\frac{\partial}{\partial p_B} s = \frac{2p_B - 1}{p_B(1 - p_B)} \cdot \frac{(p_B(1 - p_B))^r}{(p_A(1 - p_A))^r + (p_B(1 - p_B))^r} \cdot s.$$

Substituting this into the condition for $N_{00}$-monotonicity, we get

$$\frac{1}{s} \left( p_A \frac{\partial}{\partial p_A} + p_B \cdot \frac{\partial}{\partial p_B} \right) s$$
$$= \frac{2p_A - 1}{1 - p_A} \cdot \frac{(p_A(1 - p_A))^r}{(p_A(1 - p_A))^r + (p_B(1 - p_B))^r} + \frac{2p_B - 1}{1 - p_B} \cdot \frac{(p_B(1 - p_B))^r}{(p_A(1 - p_A))^r + (p_B(1 - p_B))^r}.$$

Note that the two large fractions sum to 1, so that we recognize this as the weighted average of $(2p_A - 1)/(1 - p_A)$ and $(2p_B - 1)/(1 - p_B)$, which are exactly the two terms in the maximum of the upper bound of the $N_{00}$-monotonicity condition. Furthermore, note that both these terms are larger than $-1$, so that the lower bound is also satisfied.

Similarly, for the condition corresponding to $N_{10}$-monotonicity, we get

$$\frac{1}{s} \left( (1 - p_A) \frac{\partial}{\partial p_A} - p_B \frac{\partial}{\partial p_B} \right) s$$
$$= \left( 2 - \frac{1}{p_A} \right) \frac{(p_A(1 - p_A))^r}{(p_A(1-p_A))^r + \frac{1}{2}(p_B(1-p_B))^r} + \left( 2 - \frac{1}{1-p_B} \right) \frac{(p_B(1 - p_B))^r}{(p_A(1-p_A))^r + (p_B(1-p_B))^r}.$$

Again, we recognize this as the weighted average of $2 - p_A^{-1}$ and $2 - (1 - p_B)^{-1}$, which are the terms in the minimum of the required lower bound, so that this is always satisfied. Finally, the corresponding upper bound is always satisfied since $2 - p_A^{-1}$ and $2 - (1 - p_B)^{-1}$ can both be upper-bounded by $1$. We thus conclude that $\mathrm{GM}_r$ indeed lies inside this class of measures for all $r$. $\square$

**Proof of Statement 2**   Finally, let us show that Generalized Means generalizes both CC and SBA. Recall that
$$\mathrm{GM}_r = \frac{nc_{11} - a_1 b_1}{\left( \frac{1}{2} \left( a_1^r a_0^r + b_1^r b_0^r \right) \right)^{\frac{1}{r}}}.$$

Taking $r = -1$, we obtain:

$$1 + \mathrm{GM}_{-1} = 1 + \frac{1}{2} \left( \frac{nc_{11}}{a_0 a_1} + \frac{nc_{11}}{b_0 b_1} - \frac{b_1}{a_0} - \frac{a_1}{b_0} \right)$$
$$= \frac{1}{2} \left( \frac{c_{11}(a_0 + a_1)}{a_0 a_1} + \frac{c_{11}(b_0 + b_1)}{b_0 b_1} - \frac{b_1}{a_0} - \frac{a_1}{b_0} + 2 \right)$$
$$= \frac{1}{2} \left( \frac{c_{11}}{a_1} + \frac{c_{11}}{b_1} + \frac{n - a_1 - b_1 + c_{11}}{a_0} + \frac{n - a_1 - b_1 + c_{11}}{b_0} \right)$$
$$= 2 \cdot \mathrm{SBA}.$$

Now, let us confirm that taking $r \to 0$ we get CC. Let $X := b_0 b_1 / (a_0 a_1)$, then $\left( \frac{1}{2} (a_1^r a_0^r + b_1^r b_0^r) \right)^{\frac{1}{r}}$ can be rewritten to

$$a_0 a_1 \left( \frac{1}{2} (1 + X^r) \right)^{\frac{1}{r}} = a_0 a_1 \exp \left( \frac{1}{r} \ln \left( \frac{1}{2} (1 + X^r) \right) \right).$$

We take the limit of the exponent and use l'Hôpital to find that

$$\lim_{r \to 0} \frac{\ln \left( \frac{1}{2} (1 + X^r) \right)}{r} = \lim_{r \to 0} \frac{\ln(X) X^r}{1 + X^r} = \frac{1}{2} \ln X.$$

Hence, as $r \to 0$, the denominator of $\mathrm{GM}_r$ converges to

$$a_0 a_1 \cdot \exp \left( \frac{1}{2} \ln X \right) = a_0 a_1 \sqrt{X} = \sqrt{a_0 a_1 b_0 b_1}$$

and we obtain CC.

Table 9: Examples of triplets discriminating all pairs of different measures: the upper table lists the triplets, the lower table specifies which triplet discriminates a particular pair

|  | $A$ | $B_1$ | $B_2$ |
|---|---|---|---|
| Triplet 1 | (1, 1, 1, 0, 1, 1, 0, 1, 1, 0) | (1, 1, 1, 0, 1, 0, 1, 1, 1, 1) | (1, 0, 0, 1, 0, 1, 0, 1, 1, 0) |
| Triplet 2 | (0, 1, 1, 1, 1, 0, 1, 1, 0, 1) | (1, 0, 0, 1, 0, 1, 0, 1, 1, 0) | (0, 1, 0, 0, 0, 0, 0, 0, 0, 0) |
| Triplet 3 | (0, 0, 0, 0, 1, 1, 1, 0, 1, 0) | (1, 1, 1, 1, 1, 1, 1, 1, 0, 1) | (0, 1, 1, 1, 1, 0, 1, 1, 0, 1) |
| Triplet 4 | (0, 1, 1, 1, 1, 0, 1, 1, 0, 1) | (1, 1, 1, 1, 1, 1, 1, 1, 0, 1) | (0, 1, 0, 1, 1, 1, 1, 1, 0, 1) |
| Triplet 5 | (0, 0, 0, 0, 1, 1, 1, 0, 1, 0) | (0, 1, 1, 0, 0, 1, 0, 0, 0, 1) | (0, 1, 0, 0, 0, 0, 0, 0, 0, 0) |
| Triplet 6 | (1, 1, 1, 1, 1, 1, 1, 1, 0, 1) | (1, 1, 1, 0, 1, 1, 0, 1, 1, 0) | (0, 1, 1, 0, 0, 1, 0, 0, 0, 1) |

|  | Acc | BA | $F_1$ | $\kappa$ | CE | $GM_1$ | CC | SBA |
|---|---|---|---|---|---|---|---|---|
| Acc | — | 1 | 2 | 6 | 6 | 1 | 5 | 5 |
| BA | 1 | — | 1 | 1 | 1 | 3 | 3 | 1 |
| $F_1$ | 2 | 1 | — | 2 | 2 | 1 | 2 | 2 |
| $\kappa$ | 6 | 1 | 2 | — | 4 | 1 | 3 | 3 |
| CE | 6 | 1 | 2 | 4 | — | 1 | 3 | 3 |
| $GM_1$ | 1 | 3 | 1 | 1 | 1 | — | 5 | 1 |
| CC | 5 | 3 | 2 | 3 | 3 | 5 | — | 4 |
| SBA | 5 | 1 | 2 | 3 | 3 | 1 | 4 | — |

# E  Additional experimental results

## E.1  Binary measures

**Distinguishing binary measures**  Let us show triplets of labelings $(A, B_1, B_2)$ discriminating all pairs of measures in the binary classification case. Each triplet consists of the true labeling $A$ and two predicted labelings $B_1$ and $B_2$. We say that two measures are strictly inconsistent if, according to the first one, $B_1$ is closer to $A$, while, according to the second one, $B_2$ is closer to $A$ (comparing to the main text, here we consider only strict inequalities). Table 9 lists six triplets, where all labelings are of size $n = 10$. It also specifies which triplet discriminates each pair of measures.

**Experiment within a weather forecasting service**  In this section, we provide a detailed analysis of the precipitation prediction task discussed in Section 5.1.

In Figure 1, we show the dependence of measures on the threshold that is used to convert soft predictions to binary labels. This is done separately for two prediction horizons: ten minutes and two hours. We make the following observations. For the ten-minute horizon, most of the measures agree that the optimal threshold is 0.9. However, Confusion Entropy favors the largest threshold, while Balanced Accuracy favors the smallest one. Interestingly, the behavior of measures significantly differs for the two-hour prediction interval. In this case, many of the measures favor either 0.6, 0.7, or 0.77. However, accuracy and CE prefer the largest threshold, while BA and SBA prefer the smallest one. Interestingly, this is the only experiment where we observe that SBA has such a noticeable disagreement with $GM_1$ and CC.

To better understand the differences between the measures, let us list average confusion matrices for the ten-minute and two-hour prediction horizons depending on a threshold (in increasing order). Here we show the relative values in percentages.

For ten minutes:

$$\begin{pmatrix} 93.55 & 1.12 \\ 0.22 & 5.11 \end{pmatrix} \quad \begin{pmatrix} 93.76 & 0.91 \\ 0.29 & 5.04 \end{pmatrix} \quad \begin{pmatrix} 93.84 & 0.83 \\ 0.33 & 5.01 \end{pmatrix} \quad \begin{pmatrix} 93.91 & 0.76 \\ 0.36 & 4.97 \end{pmatrix} \quad \begin{pmatrix} 94.10 & 0.57 \\ 0.49 & 4.85 \end{pmatrix} \quad \begin{pmatrix} 94.33 & 0.34 \\ 0.75 & 4.59 \end{pmatrix}$$

For two hours:

$$\begin{pmatrix} 90.41 & 4.25 \\ 1.47 & 3.87 \end{pmatrix} \quad \begin{pmatrix} 91.32 & 3.34 \\ 1.74 & 3.60 \end{pmatrix} \quad \begin{pmatrix} 91.67 & 2.99 \\ 1.87 & 3.47 \end{pmatrix} \quad \begin{pmatrix} 91.96 & 2.70 \\ 1.99 & 3.35 \end{pmatrix} \quad \begin{pmatrix} 92.94 & 1.72 \\ 2.51 & 2.83 \end{pmatrix} \quad \begin{pmatrix} 93.98 & 0.68 \\ 3.43 & 1.91 \end{pmatrix}$$

Consider, for instance, the two smallest thresholds for the ten-minute horizon. It is easy to see that accuracy grows from 98.66% to 98.80%. In contrast, for Balanced Accuracy, the difference between

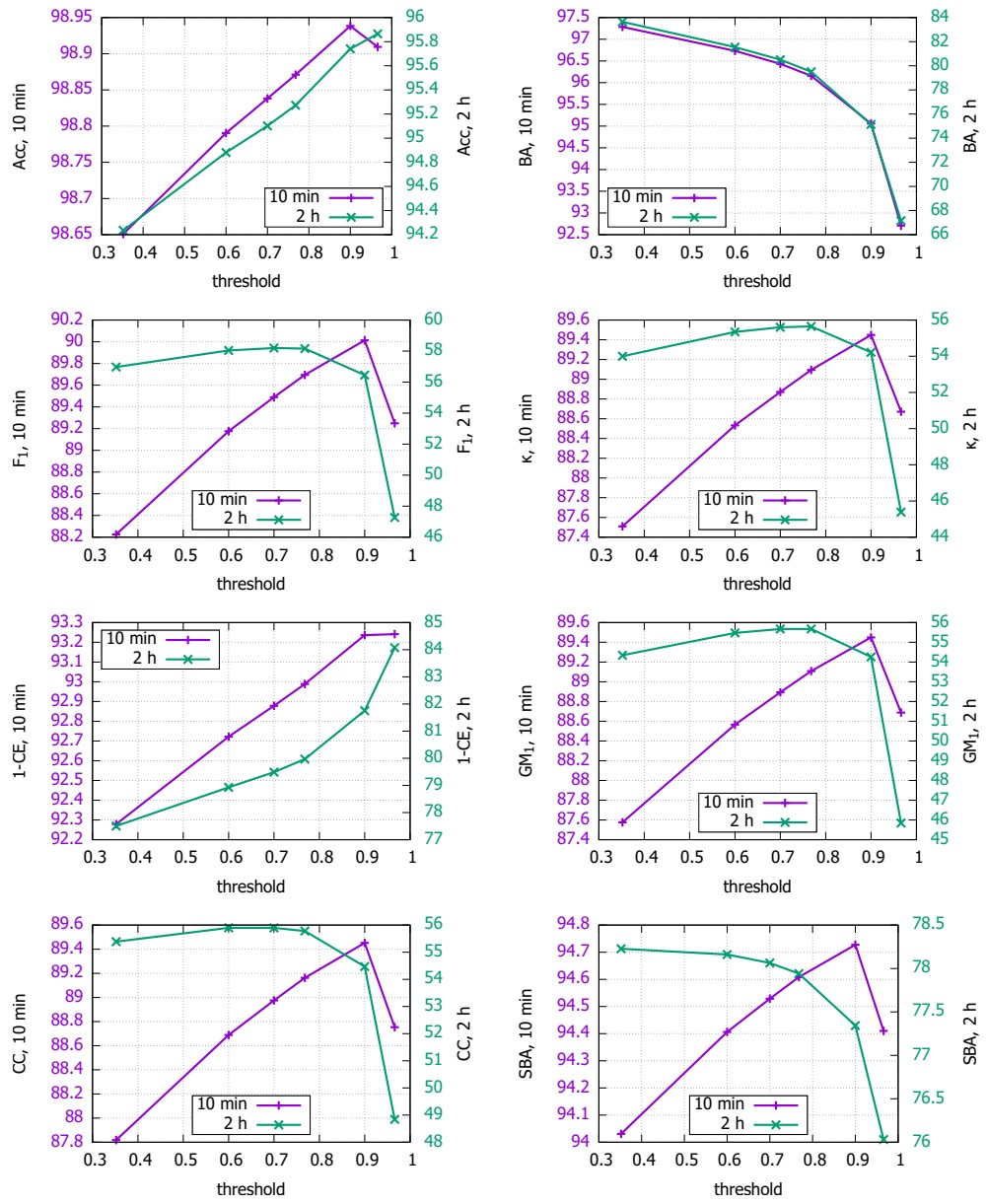

Figure 1: Dependence of measures on thresholds, for ten-minute and two-hour prediction horizons, the values are multiplied by 100

Table 10: Inconsistency of binary measures for rain prediction, horizon 10 minutes, %

|       | Acc  | BA   | $F_1$ | $\kappa$ | CE   | $GM_1$ | CC   | SBA  |
|-------|------|------|------|------|------|------|------|------|
| Acc   | —    | 93.3 | 14.4 | 14.4 | 3.3  | 14.4 | 15.0 | 15.0 |
| BA    | 93.3 | —    | 78.9 | 78.9 | 96.7 | 78.9 | 78.3 | 78.3 |
| $F_1$ | 14.4 | 78.9 | —    | 0.0  | 17.8 | 0.0  | 0.6  | 0.6  |
| $\kappa$ | 14.4 | 78.9 | 0.0  | —    | 17.8 | 0.0  | 0.6  | 0.6  |
| CE    | 3.3  | 96.7 | 17.8 | 17.8 | —    | 17.8 | 18.3 | 18.3 |
| $GM_1$ | 14.4 | 78.9 | 0.0  | 0.0  | 17.8 | —    | 0.6  | 0.6  |
| CC    | 15.0 | 78.3 | 0.6  | 0.6  | 18.3 | 0.6  | —    | 0.0  |
| SBA   | 15.0 | 78.3 | 0.6  | 0.6  | 18.3 | 0.6  | 0.0  | —    |

Table 11: Inconsistency of binary measures for rain prediction, horizon 2 hours, %

|       | Acc  | BA   | $F_1$ | $\kappa$ | CE   | $GM_1$ | CC   | SBA  |
|-------|------|------|------|------|------|------|------|------|
| Acc   | —    | 98.3 | 63.3 | 58.3 | 1.7  | 61.1 | 72.2 | 91.7 |
| BA    | 98.3 | —    | 35.0 | 39.4 | 100  | 37.2 | 25.6 | 6.1  |
| $F_1$ | 63.3 | 35.0 | —    | 4.4  | 65.0 | 2.2  | 8.9  | 28.3 |
| $\kappa$ | 58.3 | 39.4 | 4.4  | —    | 60.0 | 2.2  | 13.3 | 32.8 |
| CE    | 1.7  | 100  | 65.0 | 60.0 | —    | 62.8 | 73.9 | 93.3 |
| $GM_1$ | 61.1 | 37.2 | 2.2  | 2.2  | 62.8 | —    | 11.1 | 30.6 |
| CC    | 72.2 | 25.6 | 8.9  | 13.3 | 73.9 | 11.1 | —    | 18.9 |
| SBA   | 91.7 | 6.1  | 28.3 | 32.8 | 93.3 | 30.6 | 18.9 | —    |

the values can be written as:

$$\Delta\text{BA} = \frac{\Delta c_{00}}{a_0} + \frac{\Delta c_{11}}{a_1} \approx \frac{0.21}{94.67} + \frac{-0.07}{5.33} < 0.$$

So, Balanced Accuracy favors the smallest threshold. This can be explained by the fact that BA normalizes true positives ($c_{11}$) by a much smaller value, so that the impact of $c_{11}$ is much higher.

More interesting is the fact that for the ten-minute horizon, SBA agrees with most of the measures and strongly disagrees with BA. This can be explained by the fact that SBA also takes into account the distribution of predicted labels. For instance, for the two smallest thresholds, the difference becomes:

$$\Delta\text{SBA} \approx \frac{0.21}{94.67} + \frac{-0.07}{5.33} + \left(\frac{93.76}{94.05} - \frac{93.55}{93.77}\right) + \left(\frac{5.04}{5.95} - \frac{5.11}{6.23}\right) > 0.$$

Here the difference between the last two terms is positive and dominates all other differences. This happens because the false positive rate becomes significantly smaller.

Tables 10 and 11 summarize inconsistency between different measures for the ten-minute and two-hour horizons. In particular, we can see that SBA and CC always agree for the ten-minute horizon, while they have almost 20% disagreement for two hours.

### E.2 Multiclass measures

**Image classification**    The extended results are shown in Table 12. The models are the following:[7]

1. tf_efficientnet_l2_ns
2. tf_efficientnet_l2_ns_475
3. swin_large_patch4_window12_384
4. tf_efficientnet_b7_ns
5. tf_efficientnet_b6_ns
6. swin_base_patch4_window12_384
7. swin_large_patch4_window7_224

---

[7]https://github.com/rwightman/pytorch-image-models/blob/master/results/results-imagenet.csv

Table 12: Extended results for ImageNet, the values are multiplied by 100, inconsistencies are highlighted

|  | Acc/BA | $F_1$ | J | $\kappa$ | $1-$CE | $GM_1$ | CC | $CC^{mac}$ | SBA |
|---|---|---|---|---|---|---|---|---|---|
| 1 | 88.33 | 88.21 | 80.43 | 88.32 | 94.42 | 88.19 | 88.32 | 88.31 | 88.44 |
| 2 | 88.23 | 88.08 | 80.25 | 88.21 | 94.38 | 88.07 | 88.22 | 88.20 | 88.35 |
| 3 | 87.15 | 87.01 | 78.63 | 87.13 | 93.86 | 87.00 | 87.13 | 87.14 | 87.30 |
| 4 | 86.83 | 86.64 | 78.08 | 86.82 | 93.64 | 86.63 | 86.82 | 86.78 | 86.95 |
| 5 | 86.46 | 86.30 | 77.525 | 86.44 | 93.41 | 86.28 | 86.44 | 86.419 | 86.57 |
| 6 | 86.43 | 86.27 | **77.531** | 86.42 | **93.51** | 86.26 | 86.42 | **86.423** | **86.60** |
| 7 | 86.32 | 86.17 | 77.311 | 86.30 | 93.37 | 86.16 | 86.30 | 86.31 | 86.48 |
| 8 | 86.31 | 86.12 | **77.314** | 86.29 | **93.41** | 86.10 | 86.30 | 86.28 | 86.47 |
| 9 | 86.08 | 85.89 | 76.97 | 86.06 | 93.21 | 85.87 | 86.07 | 86.02 | 86.19 |
| 10 | 85.72 | 85.55 | 76.51 | 85.70 | 93.05 | 85.53 | 85.70 | 85.70 | 85.89 |

Table 13: Inconsistency of multiclass measures on the Yeast dataset, %

|  | Acc | BA | $F_1$ | J | $\kappa$ | CE | $GM_1$ | CC | SBA |
|---|---|---|---|---|---|---|---|---|---|
| Acc | — | 11.8 | 13.7 | 11.1 | 4.6 | 47.7 | 11.1 | 3.3 | 17.0 |
| BA | 11.8 | — | 9.8 | 8.5 | 7.2 | 52.9 | 7.2 | 8.5 | 11.8 |
| $F_1$ | 13.7 | 9.8 | — | 2.6 | 10.5 | 48.4 | 5.2 | 10.5 | 4.6 |
| J | 11.1 | 8.5 | 2.6 | — | 9.2 | 49.7 | 6.5 | 9.2 | 7.2 |
| $\kappa$ | 4.6 | 7.2 | 10.5 | 9.2 | — | 49.7 | 7.8 | 1.3 | 13.7 |
| CE | 47.7 | 52.9 | 48.4 | 49.7 | 49.7 | — | 51.0 | 48.4 | 45.1 |
| $GM_1$ | 11.1 | 7.2 | 5.2 | 6.5 | 7.8 | 51.0 | — | 7.8 | 8.5 |
| CC | 3.3 | 8.5 | 10.5 | 9.2 | 1.3 | 48.4 | 7.8 | — | 13.7 |
| SBA | 17.0 | 11.8 | 4.6 | 7.2 | 13.7 | 45.1 | 8.5 | 13.7 | — |

8. dm_nfnet_f6

9. tf_efficientnet_b5_ns

10. dm_nfnet_f5

Note that the dataset is balanced, so accuracy coincides with BA, and weighted average coincides with macro average.

**Inconsistency for Yeast dataset**  In this experiment, we consider the Yeast dataset[8] from the UCI repository [9]. The task is to predict protein localization sites among ten possible variants. The class sizes are {463, 429, 244, 163, 51, 44, 35, 30, 20, 5}, so the dataset is highly unbalanced.

To this dataset, we apply the following algorithms from the scikit-learn library [25]: DecisionTree, ExtraTree, ExtraTreesEnsemble, NearestNeighbors, RadiusNeighbors, RandomForest, BernoulliNB, GaussianNB, LabelSpreading, QuadraticDiscriminantAnalysis, LinearDiscriminantAnalysis, NearestCentroid, MLPClassifier, LogisticRegression, LogisticRegressionCV, RidgeClassifier, RidgeClassifierCV, LinearSVC. Thus, there are 18 algorithms giving 153 possible pairs. For each pair of measures and each pair of algorithms, we check whether the measures are consistent. Aggregating the results over all pairs of algorithms, we obtain Table 13.

We can see that for some measures the disagreement can be significant. For example, inconsistency is particularly high for Confusion Entropy, which does not satisfy most of the properties. Interestingly, the best agreement is achieved by CC and $\kappa$.

Finally, Table 14 shows inconsistency of different averagings.

---

[8]https://archive.ics.uci.edu/ml/datasets/Yeast

Table 14: Inconsistency of averagings on the Yeast dataset

| | $F_1^{mic}$ | $F_1^{mac}$ | $F_1^{wgt}$ |
|---|---|---|---|
| $F_1^{mic}$ | — | 13.73 | 3.27 |
| $F_1^{mac}$ | 13.73 | — | 10.46 |
| $F_1^{wgt}$ | 3.27 | 10.46 | — |

| | $J^{mic}$ | $J^{mac}$ | $J^{wgt}$ |
|---|---|---|---|
| $J^{mic}$ | — | 11.11 | 2.61 |
| $J^{mac}$ | 11.11 | — | 8.50 |
| $J^{wgt}$ | 2.61 | 8.50 | — |

| | CC | $CC^{mic}$ | $CC^{mac}$ | $CC^{wgt}$ |
|---|---|---|---|---|
| CC | — | 3.27 | 0.00 | 0.65 |
| $CC^{mic}$ | 3.27 | — | 0.00 | 0.65 |
| $CC^{mac}$ | 0.00 | 0.00 | — | 0.65 |
| $CC^{wgt}$ | 0.65 | 0.65 | 0.65 | — |

| | CD | $CD^{mic}$ | $CD^{mac}$ | $CD^{wgt}$ |
|---|---|---|---|---|
| CD | — | 3.27 | 0.00 | 0.65 |
| $CD^{mic}$ | 3.27 | — | 0.00 | 0.65 |
| $CD^{mac}$ | 0.00 | 0.00 | — | 0.65 |
| $CD^{wgt}$ | 0.65 | 0.65 | 0.65 | — |

| | $GM_1^{mic}$ | $GM_1^{mac}$ | $GM_1^{wgt}$ |
|---|---|---|---|
| $GM_1^{mic}$ | — | 11.76 | 7.19 |
| $GM_1^{mac}$ | 11.76 | — | 7.19 |
| $GM_1^{wgt}$ | 7.19 | 7.19 | — |