# OpenReview forum: "Good Classification Measures and How to Find Them"
_NeurIPS.cc/2021/Conference — NeurIPS 2021 Poster_

### Official Review · Reviewer_NwUA · 2021-07-13

**Rating:** 6
**Confidence:** 4

**Summary:**

This paper proposed a systematic approach to choosing a suitable validation index for classification tasks. The authors considered several binary and multiclass indices, including new ones. They examined the indices with five sets of properties based on the study [6,18_Supplementary]. For each index and each property, they formally proved or disproved that the property is satisfied. They also presented the impossibility theorem for classification.

**Ethical Concerns:**

No.

**Limitations And Societal Impact:**

The authors addressed the limitations in the final section.

**Main Review:**

At this stage, I consider this paper "marginally above the acceptance threshold" due to its overall contributions.

Originality:
It does a good job at studying various validation index for classification tasks with the five sets of properties. I like the part of the derivation of the impossibility theorem for classification because it is a good knowledge for our design of index. In addition, their proposed indices present better satisfactory of the properties than the other existing indices.

Quality: Good. However, my main question is about the selection of the properties. The five sets of properties were transferred from clustering in [6]. The authors are better to discuss the extra properties in classifications. For example, this paper misses the property of Robustness to Imbalance (IMB), shown in [18_Supplementary] for classification. In fact, IBM seems to be the critical property in terms of "class ratio changes (Brzezinski, 2020)" in class imbalance problems. If the paper includes this part and makes the the IBM for the new indices, I will give a much higher rating. For this reason, the word "adequately" in the Abstract is an overstatement.

Brzezinski, D., et al. "On the dynamics of classification measures for imbalanced and streaming data." IEEE transactions on neural networks and learning systems 31.8 (2019): 2868-2878.

- p2. a_i=sum from {j=1} to {m} is not suitable for c_00.

Clarity: Good.

Significance: Yes in terms of the five sets of properties, but missing the IBM.

**Time Spent Reviewing:**

12 hours

---

> ### Author Response · Authors · 2021-08-09
> **Reply to reviewer NwUA**
>
> Thank you very much for your feedback and suggestions!
>
> We would like to address the main concern regarding the Robustness to Imbalance (IMB) property of [18_Supplementary]. The general imbalance problem is addressed by our constant baseline property: it says that random classifiers should give the same score in expectation independently of class sizes (in both ground labels and predicted labels). IMB from [18_Supplementary] requires a constant classifier that classifies all elements to either the positive or the negative class to get a constant similarity score $k_1$ or $k_2$, respectively. One can see that our constant baseline thus implies IMB with $k_1=k_2$. While the definition of IMB does allow for the possibility of $k_1 \neq k_2$, their paper does not mention any index that satisfies IMB with such constants. Moreover, having $k_1 \neq k_2$ may lead to bias towards a particular class, which does not seem to be desired. We will extend our discussion of [18_Supplementary] to include this discussion of IMB. Referring to Table 6, IMB is satisfied by CC, BA, Cohen’s kappa, S&S, GM, and CD (if singularities are resolved as discussed in Supplemental B).
>
> Thank you for pointing out to (Brzezinski, 2020). This paper supports our discussion on the importance of the class imbalance problem and the constant baseline property. We will add this paper to our related work section. Additionally, following the review by WaUL, we also intend to extend the discussion of class imbalance by moving the description of our weather experiment (see Supplemental E.5) to the main text.

---

> ### Author Response · Authors · 2021-08-27
> **Looking forward to further discussions**
>
> We would like to thank you again for your suggestions; we improved the paper accordingly. Since there is about a week left to continue the discussion, we would like to know whether you have any additional comments after reading our response. We believe that our analysis of the constant baseline addresses the main concern regarding the robustness to imbalance. We look forward to further discussion and ready to give more details and analysis regarding this and other questions.

---

### Official Review · Reviewer_ybxe · 2021-07-13

**Rating:** 7
**Confidence:** 3

**Summary:**

The core question studied in this paper is how to choose a best performance measure (termed as "index" by the authors) in all situations for binary and multiclass classifications. To answer this question, the authors define several properties that are important for considerations when one wants to choose a good index. Then many common indices (performance measures) in classification tasks are analyzed to see if they satisfy each of the properties defined, including Accuracy, F-measure, and others. A impossibility theorem is proved to show that some desirable properties cannot be satisfied at the same time for a given index. With this theorem, a family of indices is proposed that satisfies all good properties except one.

**Ethical Concerns:**

No ethical issues.

**Limitations And Societal Impact:**

Yes, but the authors are encouraged to include a disclaimer/discussion on the fairness of different indices.

**Main Review:**

Overall I think this is an interesting and comprehensive study. The guidelines (properties) provided in the paper would potentially benefit practitioners who need to choose a performance measure for a given task. Out of the properties considered, the *constant baseline* is very important and useful. However, my main concern is about the novelty and significance in the machine learning community. See below for the detailed comments.

Originality.
There are two main issues.

1) Overall I feel that this paper is an incremental work and its novelty is of concern. This paper is based on a recent paper [1] (as the authors have acknowledged) that provides a systematic analysis of different indices for the clustering task. The authors transfer most of the properties defined in [1] to the binary and multiclass classification tasks. The transfer to binary tasks is trivial (as the authors have mentioned). One of the new contribution, that the authors claim, is to extend the properties and analyses to the multiclass case. However, I fail to appreciate how multiclass case is different from the clustering case (besides the setting of course), what the hurdles are when one transfers the properties and analyses to multiclass tasks, and whether new techniques are used. It would be more clear if the authors could elaborate the relation with [1].

2) The second concern is about the position of this paper in the machine learning community. Although this paper might be the first to present a comprehensive systematic study of different indices and properties in binary and multiclass classifications, many prospects have been touched more or less in earlier works.  I have read the related work section in the supplementary material, but I cannot fully recognize the position of this work, partially because much cited work is not from the usual machine learning venues, and I do not have enough information to bridge this paper with the background. Therefore, I decide to raise this concern to see if other reviewers could help better position this paper. Also, the authors should provide more background and related work from the machine learning venues.

Quality.
This work is theoretical and overall is technically sound. The claims are backed by theory and experiments.

Clarity.
This paper overall is clear, and is able to communicate the main message to the reader, though some improvements are still needed. Here are some suggestions.

1) Some terminologies need more explanations, for example, *reference* (true labels), *candidate* (predicted labels), *index* (performance measure), *validation index* (target performance measure). I do not know whether these terms are standard in other community, but it is the first time I have seen these terms. It would be helpful if the authors could use standard terminologies in machine learning or explain the terms so that this paper would be more accessible to people in the machine learning community. As I have said earlier that I think this work may be beneficial for practitioners working on machine learning, it is important to make this paper more accessible.

2) Tables need more details in the captions to help understand the content. Please consider moving some of the details from the body of the paper to the caption of a table.

Significance.
This paper is a comprehensive systematic study of different indices and properties in binary and multiclass classifications. The properties studied in this work can potentially serve as guidelines for practitioners who use machine learning to help them find a good performance measure for a given task. However, the use of some nonstandard terminologies and the lack of background and related work in machine learning can have negative effects on the significance and accessibility of this paper.

Take the above into consideration, especially my concern about the novelty, I would vote for a weak accept.

---

Other questions and comments:

1) Is index A with more properties ($P_A$) strictly better than index B with less properties ($P_B$ where $P_B \subsetneqq P_A$)? Should practitioners choose the index with as many properties as possible?

2) What might be the slide effect for an index with many properties? Is it harder to compute?

---

Typo:

Page 2, line 41, "MCC" -> "CC"

---
Citation:

[1]
Gösgens, M.M., Tikhonov, A. &amp; Prokhorenkova, L.. (2021). Systematic Analysis of Cluster Similarity Indices: How to Validate Validation Measures. <i>Proceedings of the 38th International Conference on Machine Learning</i>, in <i>Proceedings of Machine Learning Research</i> 139:3799-3808 Available from http://proceedings.mlr.press/v139/gosgens21a.html.



**Time Spent Reviewing:**

10 hours

---

> ### Author Response · Authors · 2021-08-09
> **Reply to Reviewer ybxe**
>
> Thank you very much for your feedback and suggestions!
>
> Let us elaborate more on our contribution compared to Gösgens et al., 2021. First of all, let us note that even for the binary classification, we obtain a number of novel results:
> * we formally analyze the properties of several indices not included in Gösgens et al.;
> * we develop a new class of indices satisfying all the properties except for being a distance (Generalized Means);
> * we formally prove the impossibility theorem that it is impossible to satisfy all the properties simultaneously.
>
> Another contribution is considering the multi-class problems. As we mention in the paper, a clustering can be considered a binary classification over the element pairs, where each pair is either classified as belonging to the same cluster or different clusters (this equivalence to binary classification does not require the clustering to consist of two clusters). This equivalence between clustering and binary classification does not extend to multi-class classification, which is, therefore, an entirely different problem. For the multi-class case, we need to properly extend all the properties. In addition to proving which indices satisfy which properties, we also prove whether different averagings preserve the properties.
> Regarding the theoretical aspect, let us note that all the proofs and counterexamples in Sections C and D of the supplemental are novel.
>
> Regarding the positioning of the paper, we will improve the related work section. To do this, we will more clearly describe the contexts (fields of study and application) in which the related works were written, especially those that lie outside the field of machine learning.
>
> Thank you for the comment regarding the terminology. We totally agree that making the text more accessible to the machine learning community is important. We are going to use the suggested terminology in the revised version. Table captions will also be improved.
>
> * *Is index A with more properties ($P_A$) strictly better than index B with less properties ($P_B$ where $P_B \subsetneqq P_A$)? Should practitioners choose the index with as many properties as possible?*
>
> The relative importance of each of these properties depends on the application. Our advice would be to select properties that are desirable for the given application and then select one that satisfies all of these. For instance, if the test set is unbalanced, then it is crucial to have the constant baseline property to avoid biases to particular classes. Also, based on all the feedback we’ve received, we are going to aggregate all our practical recommendations in the concluding section.
>
> * *What might be the slide effect for an index with many properties? Is it harder to compute?*
>
> In general, satisfying many properties does not necessarily lead to side effects. However, as our impossibility theorem shows, some sets of properties may be mutually exclusive. Furthermore, we see in the literature that modifying a given index to satisfy a specific property often leads to side effects. For example, Cohen's Kappa is obtained by modifying Accuracy such that the Constant Baseline is satisfied. This modification does come with the side effects that minimal agreement, distance, and strong monotonicity are no longer satisfied.

---

> > ### Comment · Reviewer_ybxe · 2021-08-23
> > **Raise my rating to 7**
> >
> > Thanks for the response. My main concern (the first one in the Originality section) was addressed and therefore I decided to raise my rating to 7.
> >
> > If the paper is accepted, the authors should improve the related work section: include more related papers in the machine learning venues and discuss the contexts for papers outside the machine learning community. Also, the notations need improvements as well to make the paper more accessible to people working in machine learning. Last but not least, as said in the response, some practical recommendations should be discussed to benefit practitioners more.

---

> > > ### Author Response · Authors · 2021-08-24
> > > **Thank you for the valuable suggestions**
> > >
> > > We will take all the suggestions into account to improve the paper.

---

### Official Review · Reviewer_WaUL · 2021-07-14

**Rating:** 7
**Confidence:** 3

**Summary:**

This paper proposes a number of desiderata for classification measures and examines common measures, such as accuracy, Matthew’s correlation coefficient, and F1 score, to show which metrics have desirable properties. Some properties include symmetry, distance interpretation, monotonicity, and a constant baseline.

**Limitations And Societal Impact:**

I have two complaints and a societal impact comment.

My primary complaint is that I would have benefitted from additional structure in the main paper. Specifically, a distinct proposition here or there would have been nice, but the exposition is otherwise reasonable to follow.

A secondary complaint is that I would have liked to have seen some more examples and discussion balance and imbalance in the data. There is a slight discussion in the ImageNet discussion, but I think there could be room for a bit more, given that class imbalance is one of the motivating factors behind the use of some metrics (e.g., the ever-annoying F1 measure).

This could likely be a paper on its own, but I think the paper could potentially have negative social impacts. The reason is essentially fairness: how is selecting a desirable classification measure in practice going to affect different groups? I certainly don’t view this as disqualifying--I just bring it up because we’re asked to think a bit more about these issues now. I just think a mention should be made that in practice one may also need to consider other properties of the measure.


**Main Review:**

Overall, the results are numerous, if not all that difficult (from my admittedly quick glance at the appendices) and the discussion is fairly thoughtful. I’m not overly familiar with research specifically on classification metrics, so it’s unclear to me how novel some of these ideas are. In the case that they are at least somewhat novel, this paper would be a nice starting point for discussions on metrics, which I feel is somewhat lacking in the community.



**Time Spent Reviewing:**

3

---

> ### Author Response · Authors · 2021-08-09
> **Reply to Reviewer WaUL**
>
> Thank you very much for your feedback and suggestions!
>
> We agree that the structure of the paper can be improved. Based on all the feedback we’ve received, we are thinking about moving the experimental part to after the theory to base the discussion on previous content and avoid references to later sections. Also, we want to improve Section 5 by formulating the main results (that are mentioned in this section and proven in Supplemental D) as separate statements. Finally, we are also going to aggregate all our practical recommendations in the concluding section.
>
> It is also a good suggestion to include a deeper discussion of imbalance. It is currently understood by the community that, e.g., a perfectly balanced ImageNet does not reflect real problems, and long-tailed learning and imbalanced classification are becoming more and more popular. We’ll add a discussion about that and include the reference to (Brzezinski, 2020) suggested by Reviewer NwUA.
> In our paper, Supplemental E.5 discusses an experiment on rain prediction, which is imbalanced since “no rain” occurs much more frequently than “rain”. We will move the discussion of this experiment to the main text.
>
> Regarding the social impact, we’ll definitely extend our answer in Checklist 1(c). We would like to emphasize that our analysis may help towards reducing certain biases in research. For instance, asymmetric indices such as Jaccard may have a bias towards a specific class. If we would select a classifier based on such an asymmetric index, it may result in outcomes that are unfair towards some classes. A similar problem occurs for indices with a bias towards the majority class (e.g., Accuracy). Thus, the bias towards the majority class could be even amplified with the poor metric selection. Thus, our work could provide some clues on how to avoid such a situation.

---

### Decision · Program_Chairs · 2021-09-27

**Decision:**

Accept (Poster)

**Comment:**

Throughout the discussion the reviewers agreed that the paper provides an interesting and comprehensive study of performence measures, and that it is of interest to the NeurIPS community. We urge the authors to revise the paper according to the comments provided by the reviewers in the rebuttal.

Thanks for submitting your work to NeurIPS!